# Selective oxidative protection leads to tissue topological changes orchestrated by macrophage during ulcerative colitis

Juan Du[1,14] ✉, Junlei Zhang[2,3,4,14], Lin Wang[2,3,4,14], Xun Wang[2,3,4,14], Yaxing Zhao[2,3,4], Jiaoying Lu[1], Tingmin Fan[5,6], Meng Niu[1], Jie Zhang[1], Fei Cheng[7], Jun Li[7], Qi Zhu[8], Daoqiang Zhang[8], Hao Pei[9], Guang Li[10], Xingguang Liang[5,6], He Huang[11], Xiaocang Cao[12,13] ✉, Xinjuan Liu[10] ✉, Wei Shao[8] ✉ & Jianpeng Sheng[2,3,4] ✉

Ulcerative colitis is a chronic inflammatory bowel disorder with cellular heterogeneity. To understand the composition and spatial changes of the ulcerative colitis ecosystem, here we use imaging mass cytometry and single-cell RNA sequencing to depict the single-cell landscape of the human colon ecosystem. We find tissue topological changes featured with macrophage disappearance reaction in the ulcerative colitis region, occurring only for tissue-resident macrophages. Reactive oxygen species levels are higher in the ulcerative colitis region, but reactive oxygen species scavenging enzyme SOD2 is barely detected in resident macrophages, resulting in distinct reactive oxygen species vulnerability for inflammatory macrophages and resident macrophages. Inflammatory macrophages replace resident macrophages and cause a spatial shift of TNF production during ulcerative colitis via a cytokine production network formed with T and B cells. Our study suggests components of a mechanism for the observed macrophage disappearance reaction of resident macrophages, providing mechanistic hints for macrophage disappearance reaction in other inflammation or infection situations.

Ulcerative colitis (UC) is a chronic idiopathic inflammatory bowel disorder of the colon featuring continuous mucosal inflammation, affecting millions of people worldwide, and its incidence has increased over recent years[1]. Tissue damage in UC patients is mainly driven by a dysregulated immune response. Various types of immune cells, such as monocytes, macrophages, and T cells, play a crucial role in UC pathogenesis[2].

Macrophage in the lesion is important for UC development and treatment. We and others have shown that macrophages in the intestinal space consist of resident and infiltrating inhabitants[3], and resident macrophages are believed to suppress the inflammation during UC[4] while infiltrating monocytes give rise to proinflammatory effector cells[5].

Pro-inflammatory cytokines, such as tumor necrosis factor-α (TNF-α) and interleukin-1β (IL-1β), are critical for disease progression as well[6]. And TNF-α neutralizing antibody treatment for UC has become favorable in the past decade[7] due to its successful application in the induction and maintenance of UC remission. Macrophage was reported to be involved in the TNF-α mediated UC[8,9]. However, many patients are still resistant to anti-TNF-α therapy[10]. Thus, comprehensive elucidation of macrophage population change during UC is important for understanding and intervention of UC.

The macrophage disappearance reaction (MDR) was first noticed in peritoneum space after response to certain stimuli such as inflammation and infection[11], but its mechanism was only illustrated partially in a recent report[12]. Zhang et al. proposed that peritoneal macrophages

A full list of affiliations appears at the end of the paper. ✉e-mail: dujuan@zju.edu.cn; caoxiaocang@ihcams.ac.cn; liuxinjuan@mail.ccmu.edu.cn; shaowei20022005@nuaa.edu.cn; shengjp@zju.edu.cn

disappeared after bacterial infection based on the factor V dependent coagulation process[12]. However, whether MDR happens in UC and the pathological consequence was yet not understood.

High-dimensional single-cell analytics such as mass cytometry (Cy-TOF) and single-cell RNA sequencing (scRNA-seq) have revealed cellular heterogeneity in UC[13–15] and have identified many important cellular subsets promoting or attenuating UC, such as CD14+ macrophage[16], IL-17+ CD8+ T cells[13], and CD26+ CD8+ T cells[17]. However, the tissue topology, such as spatial distribution and neighborhood interactions of different cells, is lost during both Cy-TOF and scRNA-seq analysis, which rely on dissociated cells. And the spatial immune network is yet to be defined. In contrast to scRNA-seq and CyTOF analyses, imaging mass cytometry (IMC) enables spatially resolved phenotyping of the UC ecosystem.

Here, we aimed to dissect the tissue UC ecosystem while preserving its architecture so that we might understand how tissue topology changes during UC progression. Using a combination of single-cell transcriptomics and proteomics methods, we wanted to resolve both cellular and spatial heterogeneity of the UC microenvironment at single-cell resolution, and provide mechanistic hints for macrophage disappearance reaction (MDR) in the UC region.

## Results

### Spatially resolved portrait of UC ecosystem

To dissect the spatial heterogeneity of the UC microenvironment while preserving the architecture of the UC ecosystem, we established a 40-marker IMC panel (39 protein markers and DNA) for the UC ecosystem (Fig. 1a, b), adapted from published protocols for intestinal tissue and cancer[18–21]. Our 40-marker IMC panel covered markers for epithelial, endothelial, stromal cells, and various immune cells; the cytokines IL-1β, TNF-α, and IL-6; the proliferation marker Ki-67; and the apoptotic marker cleaved caspase-3 (Supplementary Table S1), which allowed the highest level of multiplexing of IMC so far. Every antibody was validated by immunohistochemistry (IHC) before conjugation with heavy metals and each metal conjugated antibody was further validated by IMC (Supplementary Figs. S1–3).

We first analyzed colon tissues of healthy donors, which revealed typical structural markers, such as Collagen I and CD31, immunological markers, such as CD3 and CD20 (Fig. 1b), and cytokines, such as TNF-α and IL-1β, after staining for the 40-marker panel (Fig. 1b).

Distinct histological features, including intestinal crypts revealed by Pan-keratin staining (cyan) of epithelial cells and smooth muscle layer revealed by αSMA staining (yellow) of smooth muscle cells, were consistent with those determined by hematoxylin and eosin (H&E) staining but were resolved in greater detail (Fig. 1c).

We detected major cell types, including Pan-cytokeratin+ epithelial cells, CD31+ endothelial cells, αSMA+ smooth muscle cells, and Collagen I deposition in the extracellular matrix (Fig. 1c). Immune cell types detected included B cells (CD20+), T cells (CD3+). Notably, T cells and B cells were highly enriched in the intraepithelial space and gland surrounding area (Fig. 1d). Resident macrophages (CD163+ CD68+ CD11b^low), and inflammatory macrophages (CD68+ CD11b+)[21] could be detected mainly underneath the epithelial cells (Fig. 1e).

To characterize the cellular heterogeneity of the UC microenvironment while preserving gross structural features, formalin-fixed, paraffin-embedded (FFPE) human tissue samples from 19 healthy donors and 33 UC patients with different Mayo grading scores (Supplementary Table S2) were processed and stained with a 40-marker panel, before scanning by IMC.

We faced substantial imaging difficulties when cells were too close, leading to several imaging staining artifacts that would not be biologically anticipated, such as co-staining of Pan-cytokeratin staining with numerous immune cell markers (Fig. 1h). The imaging/staining artifacts were mainly due to the resolution limit of IMC technology[22], resulting in overlapped marker detection. Please note such limitations

existed in other groups relying on IMC as well. For example, T and B cell markers were located together in one cluster from Hartland et al.[18] Tumor marker and monocyte marker were shared in one cluster from a recent study by Karimi et al.[23] We tried to verify the same staining via mIHC (Supplementary Fig. S4a) and it seemed that such marker overlap couldn't be solved by imaging methods with a higher resolution limit (220 nm). However, such marker overlapping might just indicate close interaction of nearby cells in the UC microenvironment.

We also noticed marker CD3 could be separated with CD4 or CD8 (Fig. 1h). Please note, CD3 in activated T cells is reduced[24]. We performed CD3, CD4, and CD8 mIHC staining in healthy control and UC region, and it is clear to see CD3 staining in CD4 and CD8+ T cells in healthy control and UC could be separated (Supplementary Fig. S4b), due to lowered expression level after T cell activation. In addition, we observed overlapped CD45RA/RO staining pattern (Fig. 1h) and also in our previous IMC publication[21]. Our mIHC and Cy-TOF results indicated CD45RA/RO staining was partially overlapped (Supplementary Fig. S4c, d), consistent with other independent reports[25–27]. Thus, CD45RA/RO may not be mutually exclusive.

Before single-cell segmentation, the scanned images were subjected to preprocessing to improve image quality. Preprocessing consisted of compensation, denoise, and contrast enhancement steps. Compensation followed principles published in the previous publication[28]. In the image denoise step, we applied median filtering for noise suppression[29]. To enhance the contrast of each image, we followed the method published previously based on the linear regression model[30]. Our preprocessing improved the image quality significantly (Fig. 1f).

To segment individual cells or components in different channels of IMC images, we applied one connectivity-aware segmentation method described previously[31,32]. The expression level of markers in each cell was quantified and exported into matrix format.

After batch correction, about 70,000 cells were clustered into 17 cell meta-clusters were identified by PhenoGraph (Fig. 1g and Supplementary Fig. S5), which were then annotated based on the expression of key markers (Supplementary Fig. S6a). However, a single meta-cluster may include multiple cell components due to the close interaction of neighbor cells (Fig. 1h and Supplementary Fig. S6a). Myeloid cells, such as resident macrophages (CD11b^low CD163+), infiltrating macrophages (CD11b+ CD68+ CD14+) and neutrophil (CD15+) could be identified separately (Fig. 1h and Supplementary Fig. S6a). T cells (CD4+ or CD8+), regulatory T cells (FOXP3+) and B cells (CD20+) were also defined (Fig. 1h and Supplementary Fig. S6a). We also detected epithelial cells (Pan-Cytokeratin+), and mesenchymal cells (Vimentin+) (Fig. 1h and Supplementary Fig. S6a). Due to the resolution limit of IMC, a few clusters consisted of multiple cell types, such as infiltrating macrophage and T cell clusters, endothelial cell (CD31+), and smooth muscle cell (aSMA+) clusters. And one cluster from lineage negative cells was also identified (Fig. 1h and Supplementary Fig. S6a). Their frequencies in different patients are shown in Fig. 1h.

In summary, tissue architecture, major stromal cell types, diverse immune cell populations, and cytokine production in the human colon microenvironment could be observed with our 40-marker panel by IMC and our IMC results revealed a detailed landscape of the UC ecosystem, including epithelial cells, stromal cells, and various immune cells.

### MDR occurs only for resident macrophages during UC

Next, we wanted to characterize cellular changes during UC progression. Firstly, 19 healthy donors and 33 UC patients' samples were separated into normal and UC groups (Fig. 2a), based on their clinical examination results (Supplementary Table S2). Dendrogram and PCA analysis was plotted based on the cell cluster frequency of each patient, samples from the healthy donors were grouped together and samples from UC patients were clustered together (Fig. 2b).

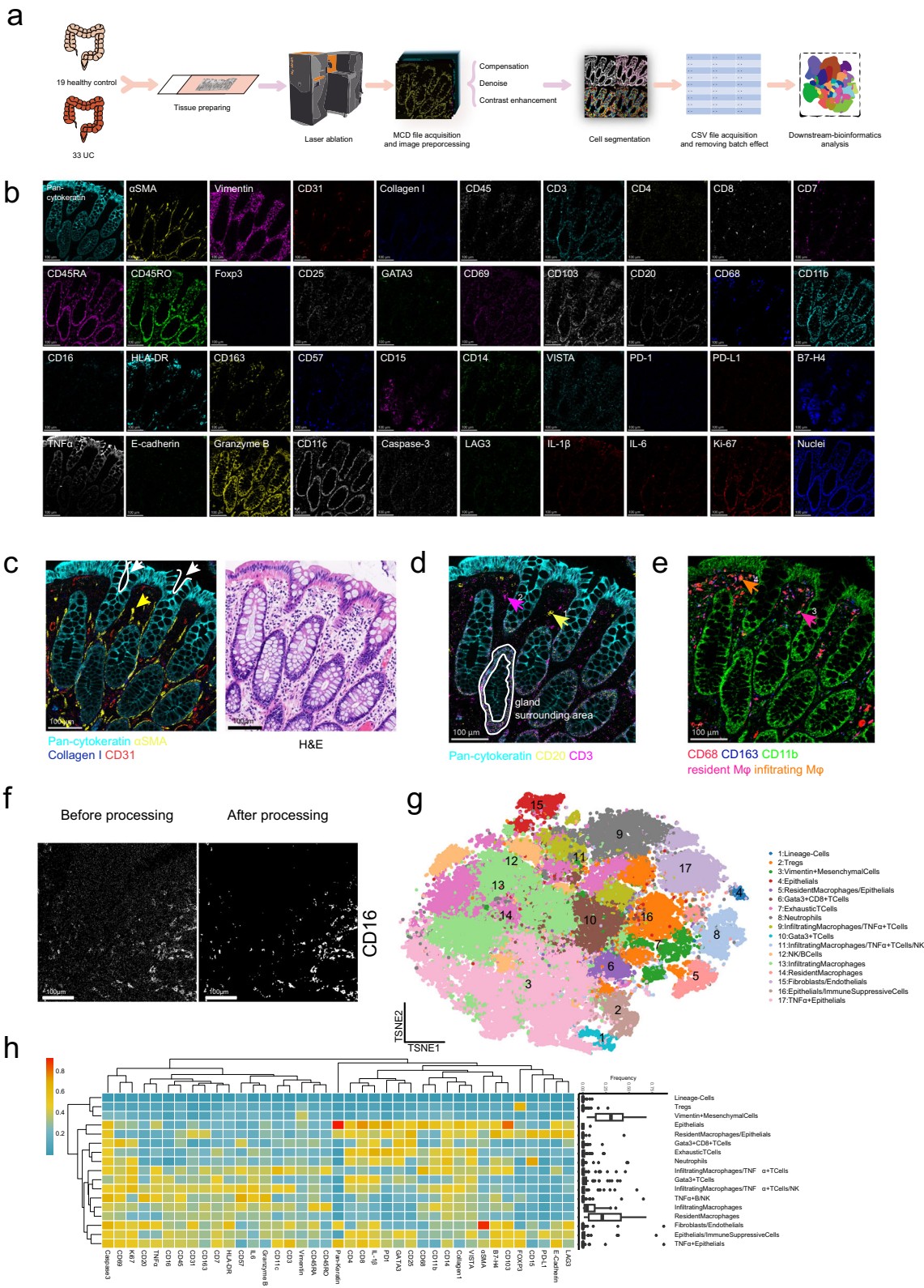

To avoid dilution effect of heavy infiltration of immune cell during UC, we compared the cell density rather than cell frequency. Although we didn't observe a statistically significant epithelial cell decrease from healthy donors to UC patients based on IMC analysis (Fig. 2c), we did notice such a trend by flow cytometry analysis of colon biospecimen (Fig. 2d). And the Vimentin⁺ mesenchymal cells also decreased (Supplementary Fig. S6b), perhaps due to increased tissue damage,

consistent with our previous publications[33–35]. In addition, T lymphocytes' meta-cluster frequency increased (Fig. 2e), indicating increased lymphocyte infiltration due to inflammation. Since the resolution of IMC for lymphocytes in the UC region was poor, above results were also validated by flow cytometry analysis. and we noticed heavy infiltration of T cells and B cells, especially for B cell and CD4⁺ T cells (Fig. 2f).

**Fig. 1 | Imaging mass cytometry pipeline based on the 40-marker panel. a** The workflow of IMC. Patients' UC and healthy donors' samples were acquired by IMC. IMC images went through preprocessing before cell segmentation, followed by batch effect removing and downstream bioinformatics analysis. **b** Single color staining of the indicated marker above each plot. Representative plot from 52 samples (2 independent experiments). **c** Pan-cytokeratin (cyan), αSMA (yellow), Collagen I (blue), and CD31 (red) were used to portray the structure of colonic tissue (left image). H&E staining (right image). Scale bars, 100 μm. The white arrows and curves point out intestinal glands. And the yellow arrows mark smooth muscles cells labeled with αSMA (yellow) in lamina propria. **d** Pan-cytokeratin (cyan), CD20 (yellow), and CD3 (magenta) highlight the distribution of T cells and B cells in colonic tissue. Arrow 1 (magenta) and arrow 2 (yellow) indicate the T cells and B

cells, respectively. Scale bars 100 μm. **e** CD68 (red), CD163 (blue) and CD11b (green) distinguish resident macrophages and inflammatory macrophages. Due to the overlap of colors, we used arrow 3 (plum) to highlight the resident macrophages (CD68+CD163+CD11b−) and arrow 4 (orange) to point out the infiltrating macrophages (CD68+CD163−CD11b+). Scale bars 100 μm. **f** Raw IMC image and processed image of CD16 staining. Scale bars, 100 μm. Representative plot from 52 samples (2 independent experiments). **g** t-SNE plots were based on the single-cell data extracted from IMC images. 17 clusters of cells from normal and ulcerative colitis samples were defined according to their markers. **h** The heat map showing the max-min normalized mean marker expression of 17 clusters with their frequency distribution pattern across different patients shown in the box plot.

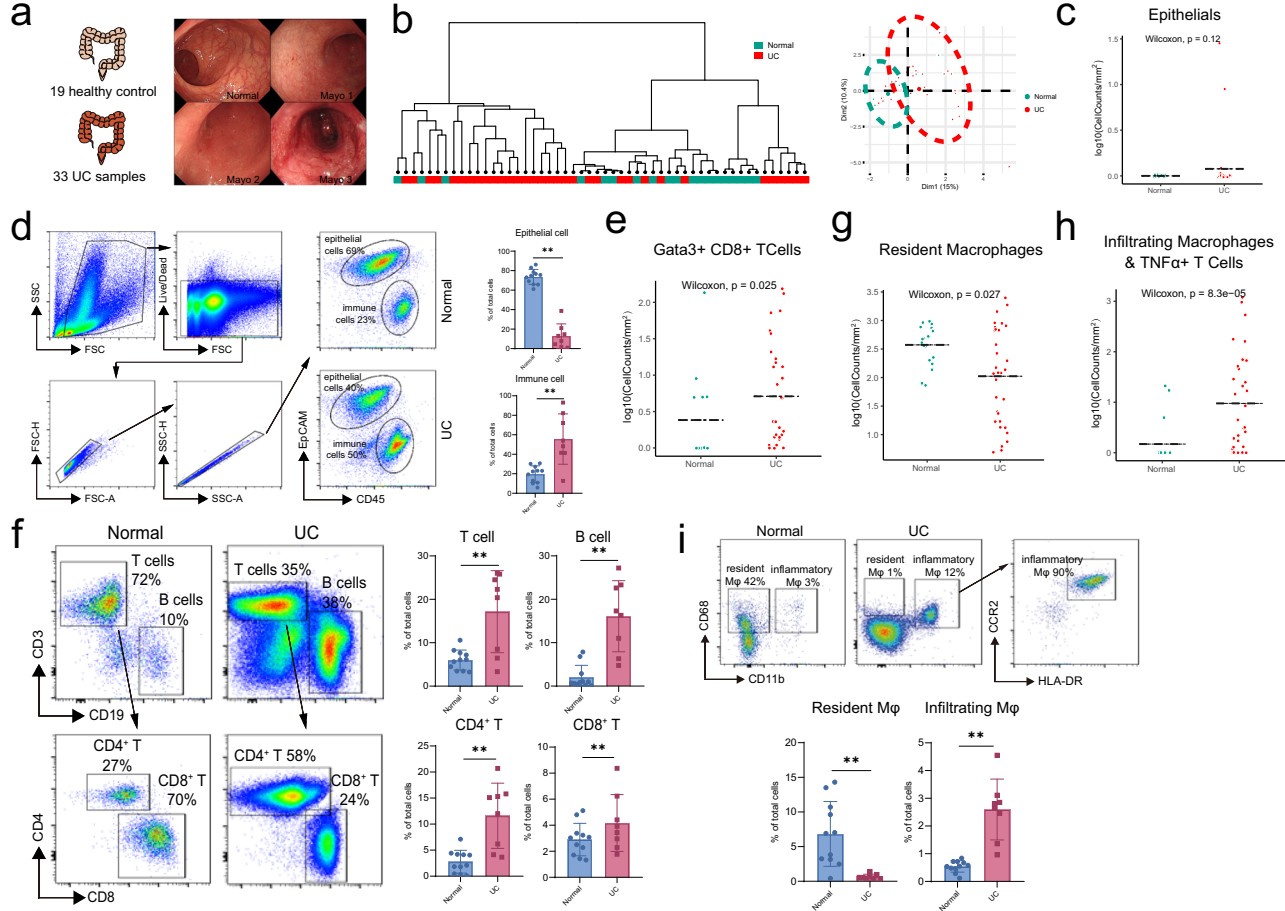

**Fig. 2 | Intestinal ecosystem changes revealed by IMC. a** 52 patient and donors' samples were separated into 2 classes: normal (green) and UC (red), determined by Mayo Clinic Endoscopic Subscore. Patients' clinical characteristics were also summarized. **b** Dendrogram based on cell type fractions showing the hierarchical relationship between samples and PCA plot showing similarities between cell type fractions of each sample from normal (green) and UC (red) groups. **c, d** The distribution of epithelial cells based on IMC results in **c**. (Normal, *n* = 19; UC, *n* = 33 from two independent experiments), statistical significance performed with two-sided Wilcoxon rank-sum tests and verification by FACS analysis of fresh samples, as shown in **d**. Statistic results (Normal, *n* = 11; UC, *n* = 8 from 8 independent experiments) were shown in the right panel of **d** as mean +/− SD and performed with a two-sided Wilcoxon rank-sum tests (***P < 0.001). Exact *P* values were provided in the Source Data file. **e, f** The distribution of T cells based on IMC results (Normal, *n* = 19; UC, *n* = 33 from two independent experiments), statistical

significance performed with two-sided Wilcoxon rank-sum tests shown in **e** and verification by FACS analysis of fresh samples, as shown in **f**. Statistic results (Normal, *n* = 11; UC, *n* = 8 from 8 independent experiments) were shown in the right panel of **f** as mean +/− SD and performed with a two-sided Wilcoxon rank-sum tests (**p < 0.01). Exact *P* values were provided in the Source Data file. **g, h** The distribution of resident macrophages and infiltrating macrophages based on IMC results (Normal, *n* = 19; UC, *n* = 33 from 2 independent experiments). Statistical significance were performed with two-sided Wilcoxon rank-sum tests. Exact *P* values were provided in the Source Data file. **i** The distribution of the resident and infiltrating macrophages by FACS analysis of fresh samples. Statistic results (Normal, *n* = 11; UC, *n* = 8 from 8 independent experiments) were shown in the right panel of **i** as mean +/− SD and performed with a two-sided Wilcoxon rank-sum tests (**P < 0.01). Exact *P* values were provided in the Source Data file.

Furthermore, we also noticed that the macrophage disappearance reaction occurred only for resident macrophages during UC (Fig. 2g). While infiltrating macrophages (Fig. 2h) increased when UC developed, suggesting a population switch occurred in the

macrophage system. Similarly, CD15+ neutrophils also increased in UC (Supplementary Fig. S6c). Macrophage disappearance reaction (MDR) was confirmed in UC patients' samples by FACS. CD68+ CD11b− resident macrophages almost disappeared in UC patients while

CD68[+]CD11b[+]CCR2[+]HLA-DR[+] infiltrating macrophages increased significantly (Fig. 2i).

Both IMC and FACS data revealed a change in cellular components with UC progression, especially for the macrophage system. MDR occurred for resident macrophages and infiltrating macrophages replaced resident macrophages.

## Macrophage replacement kinetics during UC

To analyze the macrophage replacement kinetics during UC, we established the canonical mouse DSS model and colitis score indicated by weight loss and the presence of occult blood in the feces (Fig. 3a)[4,17,36]. As published previously[3,37], CD11b[+] F4/80[hi] Fraction I cells represented colon tissue-resident macrophages, and CD11b[+] F4/80[int] Fraction II cells consisted of Ly6c[+] MHC II[−] monocytes (P1), Ly6c[+] MHC II[+] inflammatory macrophages (P2), Ly6c[−] MHC II[+] infiltrating macrophages (P3), and Ly6c[−] MHC II[−] eosinophils (P4), as described previously[3] (Fig. 3b).

Additionally, we performed a time point analysis for resident and infiltrating macrophages in an acute DSS model. 1.5% and 3% DSS treatments represented mild and severe UC (Fig. 3c). Similar macrophage replacement kinetics were observed for both the mild and severe DSS models, where resident macrophages reduced to a minimum level around day 9 before returning to a normal level around day 28 (Fig. 3c–e). Simultaneously, Ly6c[+] MHC II[+] inflammatory macrophages reached a maximum around day 9 and returned to a basal level around day 14, while Ly6c[+] MHC II[−] monocytes reached the basal level slightly later at around day 21 (Fig. 3c–e). Ly6c[−] MHC II[+] infiltrating macrophages and CD11b[+] F4/80[−] neutrophils showed different recruitment kinetics during mild and severe UC. Ly6c[−] MHC II[+] infiltrating macrophages decreased to the minimum level around day 9, similar to CD11b[+] F4/80[hi] resident macrophages, and replenishment of Ly6c[−] MHC II[+] infiltrating macrophages was completed around day 35 for mild UC but took longer for severe UC (Fig. 3c–e). Neutrophil recruitment reached a maximum between days 7–14, which was only resolved around day 28 for mild UC and much later for severe UC (Fig. 3c–e). It is well known that monocyte recruitment is dependent on the chemokine CCR2, thus we wanted to try if we could disrupt the inflammation network formation through CCR2 knockout. However, the Ly6c[+] MHC II[+] macrophage infiltration into the inflammatory region was not affected (Fig. 3f), which was probably due to the complementary roles of other chemokine receptors like CCR5.

Overall, our dynamic analysis of macrophage replacement using the mouse DSS model showed that inflammatory macrophages could replace resident macrophages during DSS treatment and return to a basal level when inflammation was resolved. In addition, CCR2 knockout was not enough to block the mouse inflammatory macrophage infiltration.

## scRNA-seq analysis confirms MDR of the UC ecosystem

To confirm the IMC pathological landscape of the UC microenvironment and illustrate the mechanism of MDR, scRNA-seq analysis was performed on four healthy donors' colons, four UC patients' self-control, and corresponding lesion samples (Fig. 4a). 42,952 single cells were captured with high sequencing quality, and we found 9 major cell populations (Fig. 4b), which was annotated based on their specific markers (Fig. 4c), such as EpCAM[+] epithelial cells, DCN[+] smooth muscle cells[38], CD31[+] endothelial cells and SPARC[+] fibroblasts[39,40] (Fig. 4c). In addition, various immune cell subsets were identified, including CD3[+] T cells, CD79B[+] B cells, CD68[+] macrophages, GNLY[+] natural killer (NK) cells[41], CPA3[+] mast cells[42] (Fig. 4c).

Refined macrophage clustering showed three subsets: resident macrophage, infiltrating macrophage, and a minor proliferating macrophage subset (Fig. 4d). CD68[+] CD163[+] CD11b[low] macrophages were featured with several resident macrophage markers, such as C1QA/B[43] and Siglec-1[3,21] (Fig. 4e). CD11b[+] CD68[+] infiltrating macrophage expressed a higher level of CCR2, which was the central chemokine for

macrophage recruitment (Fig. 4e). In addition, IL-1β expression was highly enriched in the CD11b[+] CD68[+] infiltrating macrophage subset. And Ki-67 was enriched in proliferating macrophage subset (Fig. 4e).

In addition, B cells could separated into 7 subsets (Fig. 4f), including IgM[+] naïve B cell[44], GPR183[+] B cell[45], CD27[+] IgA[+] memory B1 cells[46], CD27[+] IgG[+] memory B cells, CD69[+] CD27[+] IgA[+] resident memory B1 cells[47,48] and CD69[+] CD27[+] IgG[+] resident memory B2 cells (Fig. 4g). Similarly, further clustering of T cells showed 4 subsets (Fig. 4h), including TRGC1[+] γδT cells, Granzyme B[+] IFNG[+] CD8[+] cytotoxic T cell, LEF1[+] SELL[+] CCR7[+] naïve CD4[+] T cells[49] and FOXP3[+] CD25[+] regulatory T cells (Treg) (Fig. 4i).

Cell number for each cell subset was listed (Fig. 4j) and we also noticed infiltration of T cells and B cells in the UC region compared to the control region (Fig. 4k). And Treg cells were also elevated in the UC region (Fig. 4k), indicating negative feedback for the inflammation process in the UC region. In addition, consistent with the results shown in Fig. 2e, f, the scRNA-seq analysis showed that resident macrophages decreased, while infiltrating macrophages increased in the UC region of part of the patients (Fig. 4k), indicating the MDR occurred only in the resident macrophage subset[11].

Overall, our scRNA-seq landscape covered the major ICC TME cellular components and depicts the dynamic changes of the intestinal ecosystem, such as the MDR that occurred in the UC region, which was specific to resident macrophages, confirming the IMC results.

## Uneven expression of ROS scavenging enzyme leads to resident macrophage MDR

Next, we tried to illustrate the mechanism for resident macrophage disappearance during UC. First, we tested if MDR observed in the UC region also involved the Factor V dependent coagulation process, as previously reported[12]. However, Factor V was not expressed by resident macrophages or other cells (Fig. 5a). To explore the mechanism which resulted in MDR, we performed SCENIC analysis, a computational method for simultaneous gene regulatory network reconstruction and cell-state identification from single-cell RNA-seq data[50]. We found resident macrophages in normal and UC regions were reglated by MAF and MAFB, the key transcription factor for resident macrophages[51]. While NF-κB, the master regulator of inflammation, was specifically controlling infiltrating macrophages (Fig. 5b). SCENIC analysis results were consistent with the macrophage cell annotation.

Next, we analyzed the differentially expressed gene (DEG) between resident macrophage and infiltrating macrophage in the UC region and performed gene ontology analysis (Fig. 5c, d), and we found TNF-α signaling and inflammation pathway was upregulated in the infiltrating macrophages (Fig. 5d), consistent with NF-κB enrichment discovered in infiltrating macrophage by SCENIC analysis (Fig. 5b). Furthermore, we found ROS scavenging enzyme SOD1/2 expression was uneven. Expression level of SOD1/2 were extracted from scRNA-seq datasets for both infiltrating macrophages and resident macrophages. SOD1 was generally expressed by resident and infiltrating macrophages while SOD2 was highly expressed in infiltrating macrophages, especially in the UC region (Fig. 5e). To illustrate the consequence of uneven SOD2 expression, we performed ROS probe staining for macrophage subses via FACS. As a result, ROS level was quite high in resident macrophages compared with infiltrating macrophages, detected by FACS compatible ROS probes (Fig. 5f). ROS probe staining was performed with immunofuroscence staining stimutaneously in human samples also proved that CD169 resident macrophages were diminished in the high ROS region (Fig. 5g). We noticed opposite distribution trend of ROS and resident macrophages. There were less resident macrophages (CD169[+]) in regions with more ROS (Fig. 5h–i).

In general, we found elevated ROS levels in the UC region compared with healthy control and self-control. Meanwhile, ROS scavenging enzyme SOD2 were highly expressed only in infiltrating macrophage but not resident macrophage, which resulted ROS induced resident macrophage disbalance.

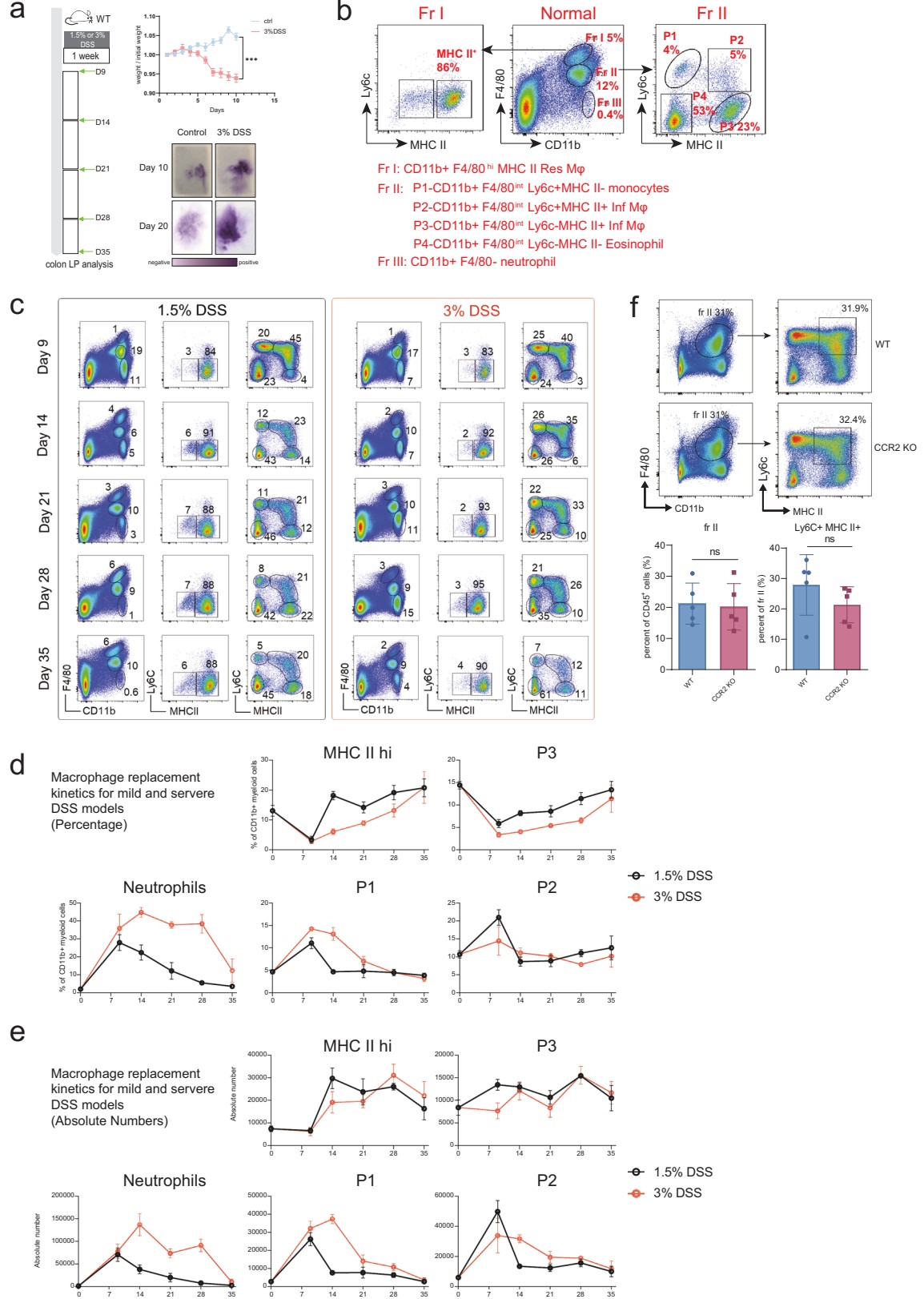

**Infiltrating macrophages orchestrate the major cellular neighborhood changes in UC regions**

It is widely accepted that the changes of macrophages can further influence the nearby immune system in situ[52]. Hence, it is critical to identify the local changes in the inflammation network following the switch of macrophages. Therefore, we performed the regional cellular neighborhood analysis. Cell neighborhood was defined as the nearest 20 neighbors to the center cell (Fig. 6a). To figure out different cellular neighborhood function units, cellular neighborhoods were annotated according to the major cellular clusters (Fig. 6b). Then based on the CN composition, a Voronoi plot was used for each IMC image to visualize whether the defined CN function units were associated with the

**Fig. 3 | Dynamics of myeloid cell changes during UC. a** DSS treatment schedule. Mice were treated with 1.5% or 3% DSS in drinking water, followed by a four-week chase period with normal drinking water. Colon tissue was isolated and analyzed at indicated time points. Weight loss and the presence of occult blood in the feces were used as indicators for the successful establishment of DSS-induced colitis model.Weight data (ctrl, $n = 5$; 3% DSS, $n = 5$ from one experiment) were shown as mean +/− SD and performed with two-way ANOVA test (***$P < 0.001$). Exact $P$ values were provided in the Source Data file. **b** Gating strategy for different myeloid cells in normal colon tissue at day 0. CD11b[+] F4/80[hi] Fraction I cells were separated into MHC II[−] and MHC II[+] subsets of resident macrophages. CD11b[+] F4/80[int] Fraction II cells were separated into P1 (Ly6c[+] MHC II[−] monocytes), P2 (Ly6c[+] MHC II[+]

inflammatory macrophage), P3 (Ly6c[−] MHC II[+] infiltrating macrophage), and P4 (Ly6c[−] MHC II[−] eosinophil). CD11b[+] F4/80[−] Fraction I cells were neutrophils. **c−e** Representative FACS plots, subset percentages, and absolute numbers for the different myeloid cells shown in Fig. 5B during different time points of the DSS model. Data (1.5% DSS, $n = 5$; 3% DSS, $n = 5$ from one experiment) were shown as mean +/− SD. **f** Representative FACS plots and statistic results of infiltrating macrophages in WT ($n = 5$) and CCR2 knockout mice ($n = 5$ from one experiment) after DSS-induced colitis. Data were shown as mean +/− SD and performed with two-tailed T test (ns $P > 0.05$, *$P < 0.05$, **$P < 0.01$). Exact $P$ values were provided in the Source Data file.

inflammation caused by UC (Fig. 6c)[53]. As shown in Fig. 6c, the topology map of IMC represented by the Voronoi plot with various CNs indicated by different colors was aligned nicely with the IMC image. We found T-enriched CN (CN5, CN6, CN10), macrophage-centered CN (CN4), and immunosuppressive CN (CN8) were higher in UC patients. While resident macrophage and mesenchymal cell-enriched CN (CN14) were more in normal samples (Fig. 6d).

Tensor analysis was performed to provide insights into how the microenvironment differs between normal and UC patients (Fig. 6e)[54]. CN-CT composition for each sample was a 2D matrix and the collection of 2D matrices from all samples formed a 3D tensor. Then tensor was decomposed in each patient group separately. After tensor decomposition, we identified CN modules and CT modules, revealing differences in the organization of the microenvironment between patient groups[54] (Fig. 6e). Different tissue modules were shown as pairs of CN and CT modules (dashed outer rectangles) interacting at different extents (indicated by the weight of the edge connecting them). We could observe infiltrating macrophage and infiltrating macrophages and T cell-organized CN (CN3, 4, and 5) and CT modules were linked with the UC sample group. While resident macrophage and mesenchymal cell-organized CN and CT modules were linked with the normal sample group (Fig. 6e).

Thus, CN changes during UC were consistent with the cellular cluster changes revealed by FACS and scRNA-seq analysis. And infiltrating macrophages orchestrated the major CN changes in UC regions.

### Infiltrating macrophages induce a spatial shift of TNF-α production

Next, we intended to explore the mechanism of UC and illustrate the functional consequences of the CN changes after resident macrophage MDR. We performed regional correlation analysis to investigate the potential spatial co-occurrence patterns of different cells across UC and donor samples and identifie statistically significant interaction or avoidance pairs within CN (Fig. 7a)[55]. First, we could observe much more interaction and avoidance pairs in UC samples than normal samples. And infiltrating macrophages were interacting frequently with TNF-α producing T, B, and NK cells (orange square 1) while showing avoidance pattern with TNF-α non-producing cells (blue square 2) in the UC samples (Fig. 7b). We also observed TNF-α non-producing cells were interacting frequently with immunosuppressive cells, implying the potential feedback mechanism of TNF-a production (Fig. 7b, orange square 3). In normal samples, resident macrophages showed avoidance and interaction pattern with TNF-α non-producing cells and Tregs, respectively. The co-occurrence pattern of resident macrophages was opposite in normal and UC samples (Fig. 7b, blue square 4), indicating distinct regulatory roles of resident macrophages in normal and UC samples.

Next, we performed cell-cell interaction analysis based on the published tool, NicheNET, by Browaeys et al.[56], which models intercellular communication by linking ligands to target genes, to identify the key regulators of the spatial interaction pattern observed. We found IL-1β from macrophages was the key driver of the TNF-α

production network with T and B cells (Fig. 7c, d). scRNA-seq helped to identify the T cells, B cells, and infiltrating macrophages as major TNF-α producers (Fig. 7e). In addition, our IMC and scRNA-seq data showed that IL-1β was produced mainly by infiltrating macrophages (Fig. 7e).

To prove the crucial role of the infiltrating macrophage in the formation of inflammation network, we sorted the Ly6c[+]MHC II[+] macrophage from DSS treated mouse and co-cultured with the lymphocytes from the draining mesenteric lymph nodes (Fig. 7f). And we found co-culture with Ly6c[+]MHC II[+] macrophage could induce both T cells and B cells to secrete TNF-α (Fig. 7f). Intriguingly, we found the spatial pattern of TNF-α production shifted in the UC region. TNF-α was mainly detected in the tropical epithelial region in the healthy colon while it was mainly detected in the lymphocytes within the lamina propria (LP) during UC.

In summary, by combining IMC and scRNA-seq data, we suggested that during UC, ROS production led to resident macrophage-specific MDR and infiltration of inflammatory macrophages, which formed the inflammatory CN with lymphocytes and orchestrated the spatial shift of TNF-α production (Fig. 7h).

## Discussion

The UC microenvironment is composed of epithelial cells, immune cells, and stromal cells. Recently, scRNA-seq has helped resolve the heterogeneity of the UC microenvironment among different patients at the transcriptional level[16,17,57]. However, single-cell spatial heterogeneity of the UC microenvironment remains to be deciphered. Here, we combined scRNA-seq analysis and IMC with a 40-marker panel to analyze UC heterogeneity in great detail, while preserving the architecture of the microenvironment. Furthermore, we discovered that MDR occurred only for resident macrophages while infiltrating macrophages replaced resident macrophages, forming the cytokine network of TNF-α and IL-1β with T cells, B cells, and NK cells.

Macrophage disappearance reaction (MDR) was first reported in the peritoneum in response to certain stimuli[11] more than two decades ago. During our continued study focusing on the macrophage, we found that MDR is a general phenomenon in many infection and inflammation conditions. We and our collaborators could observe MDR in the lung, kidney, liver, and colon in response to virus, fungi, malaria, and chemical-induced colitis[5,58–61]. Thus, we believe MDR is a true and general phenomenon in response to different stimuli but not due to the side effect of heavy infiltration of immune cells. MDR is not so well characterized, partially due to the complexity of macrophage subsets. Most tissues are comprised of both resident and infiltrating macrophages and it is important to note the MDR occurred only for the tissue-resident macrophage across tissues, such as the alveolar macrophage in the lung, F4/80[hi] resident macrophage in the kidney, Kupffer cell in liver and CX3CR1[hi] resident macrophage in colon[3]. And it is critical to distinguish resident and infiltrating macrophages well, before a detailed study of MDR.

We showed that the MDR occurred in both human and mouse UC for tissue-resident macrophages, respectively. In addition, resident macrophages only returned to normal levels when inflammation was

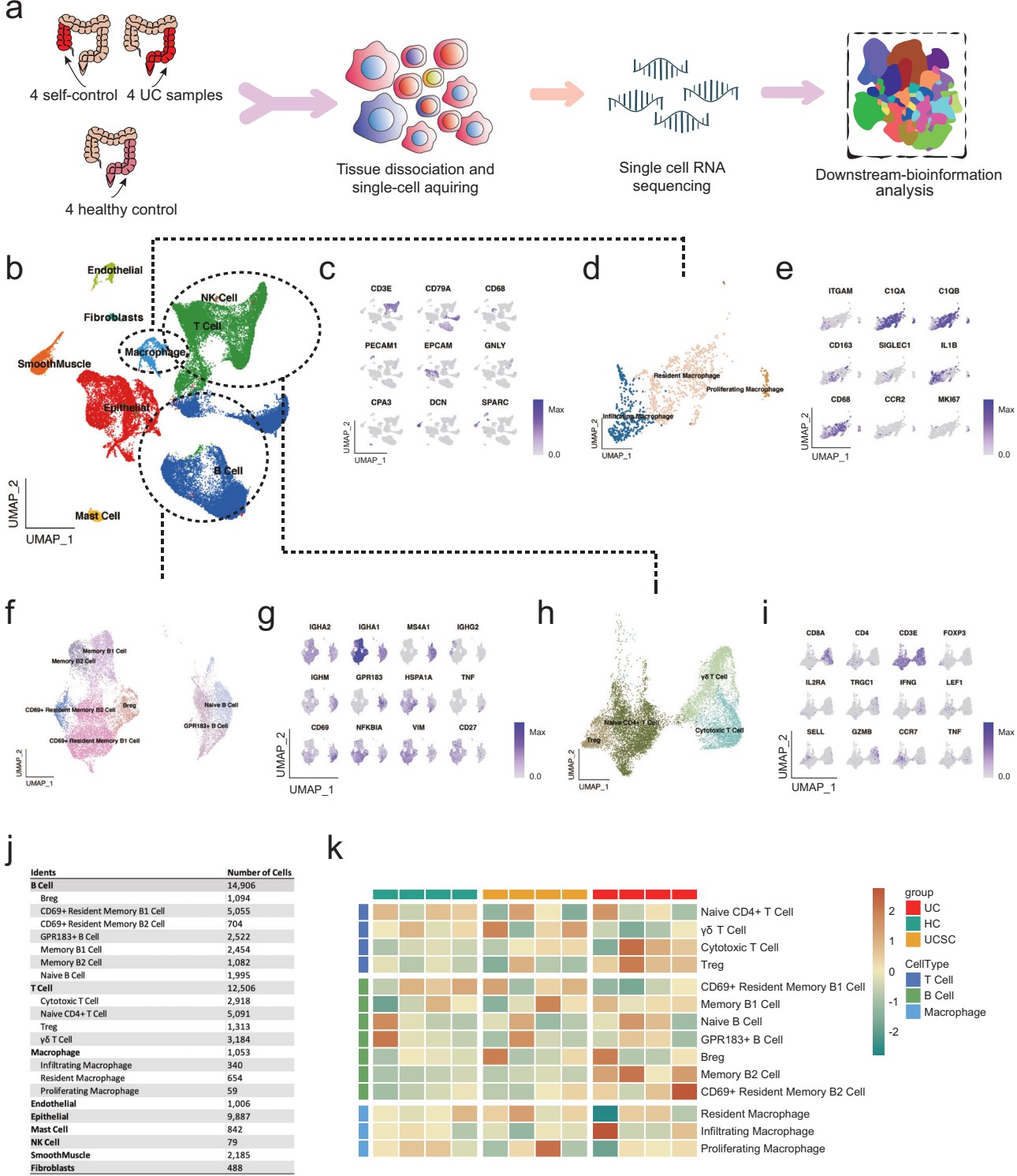

**Fig. 4 | A single-cell transcriptomic atlas of the microenvironment in UC. a** The brief schematic of the single-cell RNA-seq analysis pipeline. **b** UMAP plot showing nine major cell type clusters identified based on the scRNA-seq data. **c** UMAP plot displaying key makers of each major cell type. **d**, **e** UMAP plot displaying sub-clusters of macrophage populations and **e** Key markers of each macrophage sub-cluster. **f**, **g** UMAP plot displaying sub-clusters of B cell populations and **g** Key markers of each B cell sub-cluster. **h**, **i** UMAP plot displaying sub-clusters of T cell populations and **i** Key markers of each T cell sub-cluster. **j** Cell number statistics for major cell types and corresponding sub-clusters. **k** Heatmap showing the abundance of sub-clusters of macrophages, T, and B cells across samples. Subset frequency was normalized to total cells and row scaled by z-score.

resolved in the mouse DSS model. Thus, chronic inflammation might prevent the replenishment of resident macrophages, the major immune suppressive cells, in UC lesions of patients. Our findings were consistent with a previous report, which showed that CD14⁺ macrophages in Crohn's disease consisted of IL-1β producing CD163⁻/dim cells[16]. And CD14⁺ macrophages (likely to be infiltrating macrophages), T cells and B cells were observed to be higher in intestinal inflamed region of humans via Cy-TOF analysis[62]. In addition, our observation

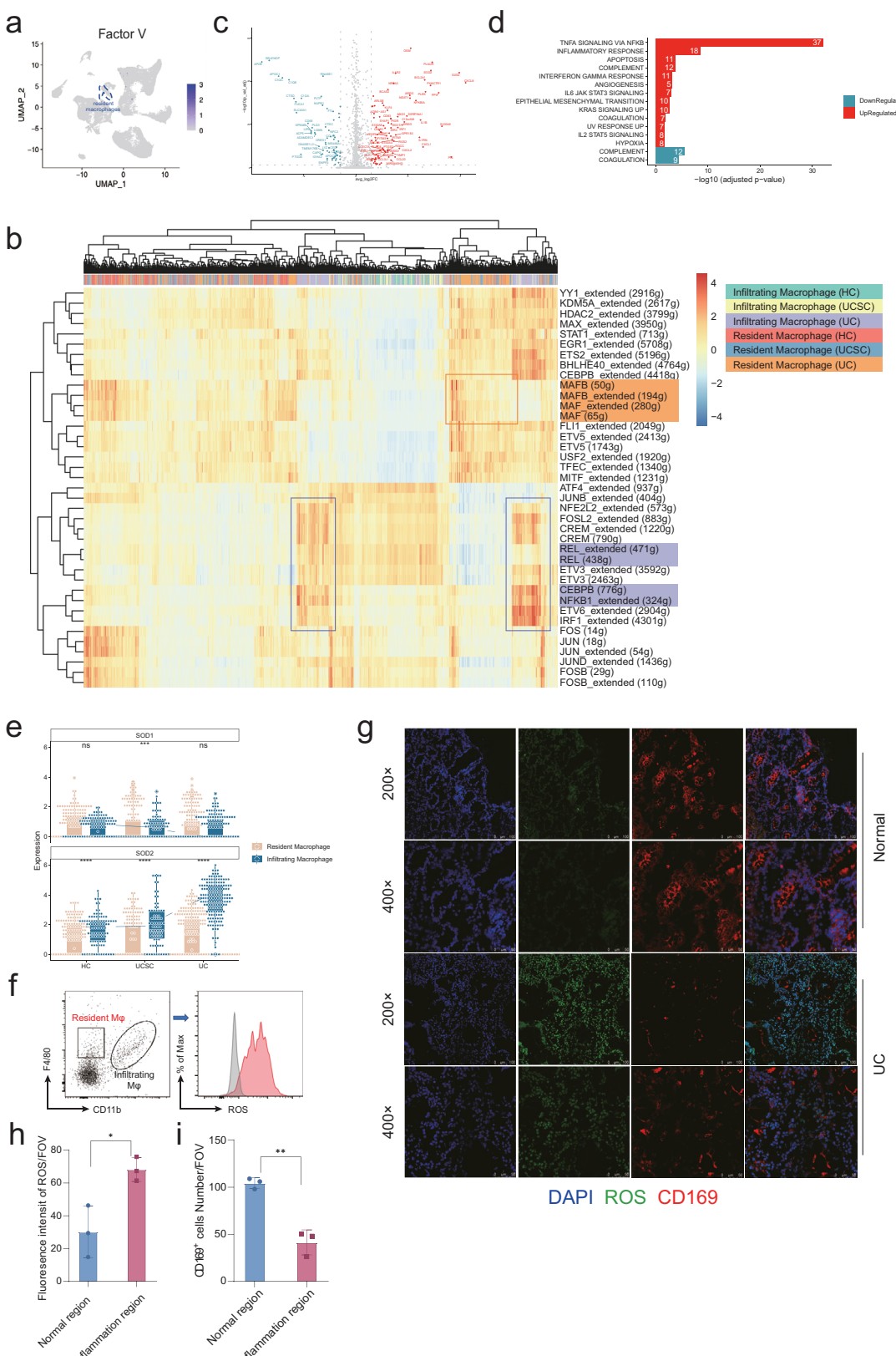

was also consistent with a recent IMC analysis of UC[63], Ayano et al. claimed tissue-resident macrophages and T Cells exhibit increased inflammatory activity in the lamina propria of patients with inflammatory bowel disease based on a 23-marker panel. We believed the resident macrophage described by Ayano et al. was infiltrating macrophages, since only CD68 was used in their annotation, and we

have clearly shown that CD11b⁺ CD68⁺ infiltrating macrophage dominated the LP region during UC.

Macrophage subsets switch dynamics were likely to be adopted as the combinatory microbe defense mechanism that ROS was elevated as the first defense of the invading bacteria. At the same time, immune-suppressive resident macrophage was wiped out purposely due to

**Fig. 5 | The mechanism of resident macrophage disappearance reaction.**
**a** UMAP identifying expression of Factor V within all major populations. **b** Heatmap of regulon activities analyzed by SCENIC for inflammatory and resident macrophages. The top row refers to cell types and sample origins. **c** Volcano plot showing genes differentially expressed between inflammatory and resident macrophages in UC patients (adjusted $p$-value < 0.05 and |log2 Fold Change| > 1.5). **d** Enrichment analysis on Hallmark gene sets based on the up-/downregulated genes compared inflammatory to resident macrophages in UC patients (adjusted $p$-value < 0.05). **e** Expressions of SOD1/2 for inflammatory and resident macrophages in HC (Resident Macrophag, $n = 199$; Infiltrating Macrophage, $n = 96$), UCSC (Resident Macrophag, $n = 201$; Infiltrating Macrophage, $n = 76$), and UC groups (Resident Macrophag, $n = 254$; Infiltrating Macrophage, $n = 168$). Each box represents the 25th to 75th percentile of values. Minima and maxima are present in the boxplot's lower and upper bounds and whiskers represent $1.5 \times IQR$ away from upper/lower quartile or maxima/minima, whichever is closer. Statistics were performed with two-sided Wilcoxon rank-sum tests (ns $P > 0.05$, *$P < 0.05$, **$P < 0.01$, ***$P < 0.001$, ****$P < 0.0001$). The scRNA-seq data were preprared from 8 experiments. Exact $P$ values were provided in the Source Data file. **f**–**i** Representative plot for ROS reporter analyzed by FACS for infiltrating and resident macrophage shown in **f** and statistical results shown in **h** ($n = 3$ in both normal and inflammation region from one experiment). Representative image for ROS reporter together with immunofluorescence staining of CD169 (a marker for resident macrophage) were shown in **g** and statistical results ($n = 3$ in both normal and inflammation region from one experiment) for the absolute count of resident macrophage shown in **i**. Data were shown as mean +/− SD and performed with two-tailed T test (*$P < 0.05$, **$P < 0.01$). Exact $P$ values were provided in the Source Data file.

their vulnerability to ROS and replaced by inflammatory macrophage which was resistant to ROS based on the high level of SOD2. However, if the inflammation process was not resolved properly, persisted loss of resident macrophages and infiltration of inflammatory macrophages would exacerbate UC. Such dysregulation of macrophage subsets was also revealed by a large-scale scRNA-seq analysis of COVID-19 patients' samples[64].

The resolution of IMC was limited for the regions with a high density of cells, resulting in overlapped marker detection, such an artifact was carried through the subsequent analysis. In fact, such an artifact can't be improved by increasing resolution limit. Marker overlapping also exists in mIHC imaging with a much higher resolution limit. However, such overlapping might just indicate the close interaction of neighbor cells and won't hamper the conclusion drawn in our manuscript. In addition, single-cell proteomics and transcriptomics analysis did complement each other. For example, when T cell and B cell markers were overlapped in areas of high cell density, scRNA-seq could resolve T cells and B cells effectively. Also, scRNA-seq was unable to determine the location of cytokine production while IMC revealed that TNF-α producing lymphocytes were surrounding the inflammatory macrophage but not resident macrophages.

We also noticed lack of CD3 co-staining with CD4 or CD8 in some of the T cells in both IMC and mIHC stainings.We believed such phenomenon was not technical artifacts, since we could observe CD3 co-staining with CD4 or CD8 in the same slide. T cells lose CD3 staining after activation and internalization of CD3 complex[24]. In addition, we noticed CD45RA/RO co-staining pattern in our previous IMC publication on HCC[21]. It seems that CD45RA/RO staining is quite overlapped also from independent reports[25–27]. Also, in our Cy-TOF analysis and mIHC staining, we could observe co-staining of CD45RA/RO. Thus, CD45RA/RO may not be so mutually exclusive.

In summary, we not only provide spatial insights into the ecosystem of UC but also identify oxidative stress as the key player responsible for the UC macrophage subset disbalance during UC. And we believe resident macrophage MDR induced by ROS was not restricted to UC, but a general phenomenon occurred in other infection and inflammation conditions, such as in the SARS-Cov-2 infection and our observation might be important for future ROS and macrophage modulation therapy for infection and inflammation diseases. Future work emphasizing the macrophage subpopulation switch due to the resident macrophage-specific MDR will be carried out to cement these observations further and elucidate the mechanism, consequence of such macrophage switch.

## Methods
### Ethics statement
The study followed the Declaration of Helsinki principles and was approved by the Medical Ethics Committee of the First Affiliated Hospital, Zhejiang University, Hangzhou, China; Tianjin Medical University General Hospital, Tianjin, China; Beijing Chaoyang Hospital, Beijing, China. Biopsy specimens from healthy volunteers and UC patients. Biopsy specimens were collected into RPMI 1640 medium on ice and processed immediately with informed consent. Animal experiments were approved by the Animal Care and Medical Ethics Committee of the First Affiliated Hospital, Zhejiang University.

### Animal studies
C57BL/6J mice were purchased from the Model Animal Research Center of Nanjing University (China). CCR2 Knockout mice were bred in-house. Only male mice were used for the mouse colitis experiment. All mice were bred in an SPF facility under the regulation of the Institutional Animal Care & Use Committee (IACUC).

### DSS-induced colitis model
Male mice at 4–6 weeks of age were randomly divided into two groups which were oral administration with 1.5 % or 3% (w/v) Dextran sulfate sodium (DSS; Sigma, 42867) dissolved in drinking water respectively for one week, followed by normal water for a four-week recovery phase. Changes in body weight were monitored every day to identify the success of the mice model throughout the experimental period. All mice were sacrificed at indicated time points, and the entire colon was collected for immunohistochemistry analysis and Fluorescence-activated cell sorting (FACS) analysis.

### Multiplex IHC
Multiplex staining of primary tissues was performed by using Opal™ 7-color multiplex IHC kit (Akoya Biosciences, NEL861001KT) according to the manufacturer's instruction[65]. Briefly, the slides were baked for approximately 1 h at 68 °C followed by de-paraffinization and rehydration. For each staining cycle, the slides were treated with the retrieval of antigen, blocking, and primary antibodies incubation, followed by HRP-conjugated secondary antibody incubation and Opal tyramide signal generation (Detailed information of antibodies and corresponding Opal fluorophores were provided in Supplementary Table S1). Then the slides were stripped with a retrieval solution as required before the next round of staining. The above process was repeated until all markers were completed. Once all markers were labeled, slides were counterstained with DAPI (Akoya Biosciences) and scanned using the PerkinElmer Vectra3® Polaris™ platform. The multispectral images obtained were unmixed using the inForm Advanced Image Analysis software (inForm 2.4.1; Akoya Biosciences, USA).

### ROS staining
For ROS staining analyzed by FACS, cell suspensions were stained first with Cell ROX Green (C10492, Thermo Fisher) for 1 h at 37 degrees, followed by antibody staining. For ROS imaging, tissues embedded in OCT (4583, Sakura) were first cut and fixed by 4% PFA for 30 min, then incubated with Cell ROX Green, followed by the primary antibody (CD169, SCBT, HSn 7D2) staining at 4 degrees overnight. Secondary staining by Anti-mouse IgG (H + L), F(ab')2 Fragment (Alexa Fluor 555 Conjugate, CST 4409) was performed the next day. Imaging acquisition was performed with LEICA TCS SP8.

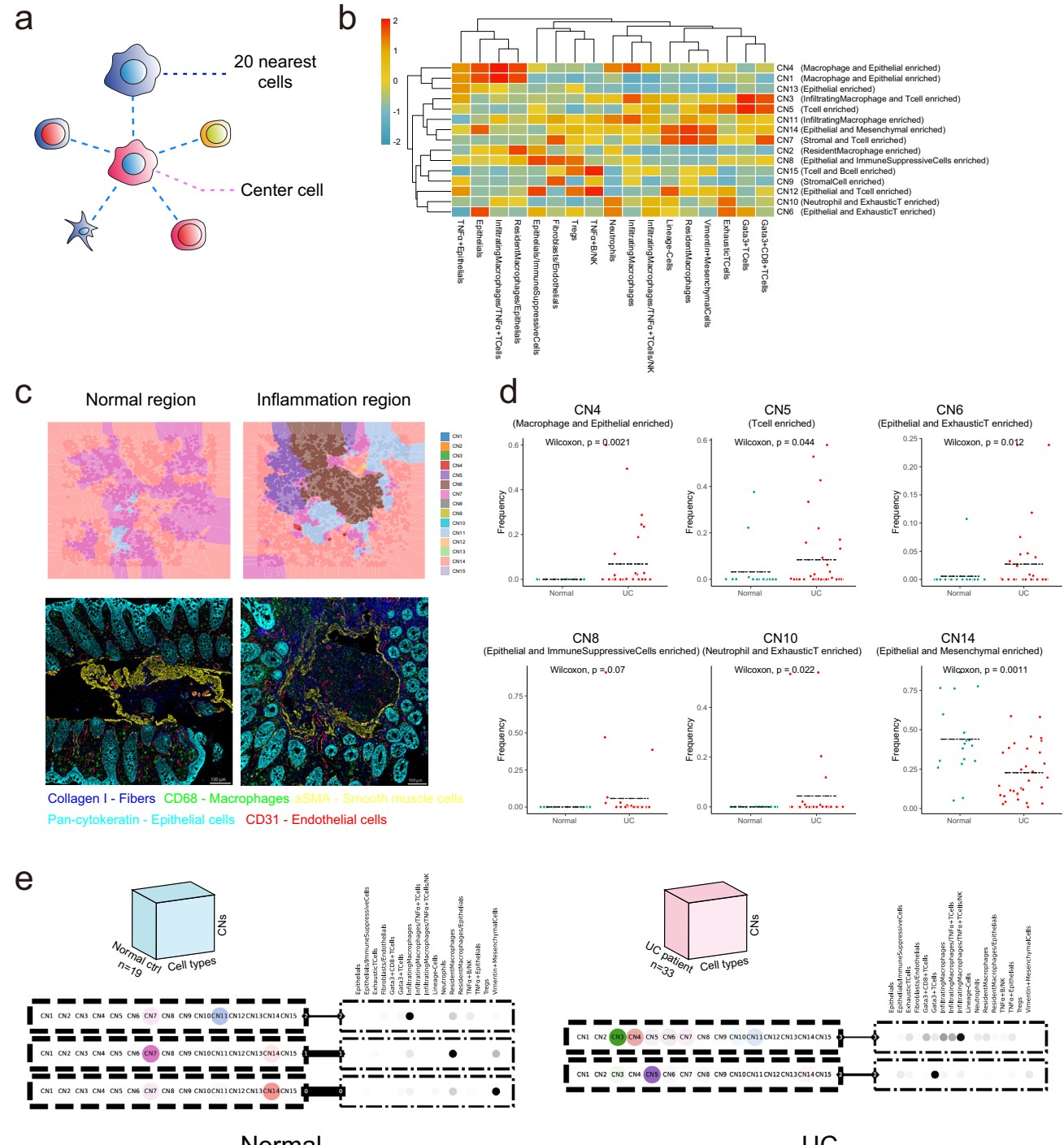

**Fig. 6 | Cellular neighborhood (CN) changes during UC. a** Schematic of CN identification. CN was defined by its center cell and the 20 nearest neighbor cells. **b** Identification of 15 distinct CNs based on the 17 original CTs and their respective abundances (column scaled) within each CN. **c** Representative Voronoi diagrams of CNs (upper panels) and corresponding IMC images (lower panels) in the normal and inflammatory region. **d** Dot plots showing the abundance of selected CNs (CN4, CN5, CN6, CN8, CN10, CN14) between normal and UC samples (Normal, *n* = 19; UC,

*n* = 33 from two independent experiments). Dashed black line labeling mean value of each group and statistics were performed with two-sided Wilcoxon rank-sum tests. Exact *P* values were provided in the plot and Source Data file. **e** Schematic of the tensor decomposition analysis and decomposition results of selected modules for normal and UC groups. Weight of the line connecting CN and CT modules indicating interaction potentials between each pair.

## IMC acquisition

Paraffin-embedded and formalin-fixed tissues from UC patients by biopsies were cut into 4 μm sections followed by heating at 68 °C for 1 h. Dewaxing in xylene was performed twice for 10 min each. Sections were rehydrated sequentially in 95%, 85%, and 75% ethanol for 5 min each, followed by 100 °C heat-mediated antigen retrieval for 30 min. After cooling naturally, sections were washed twice for 5 min each with PBS-TB (PBS with 0.5% Tween-20 (93773, Sigma-Aldrich) and 1% Bovine Serum Albumin (SRE0098, Sigma-Aldrich). Then, sections were blocked with SuperBlock (37515, Thermo Fisher Scientific) for 30 min.

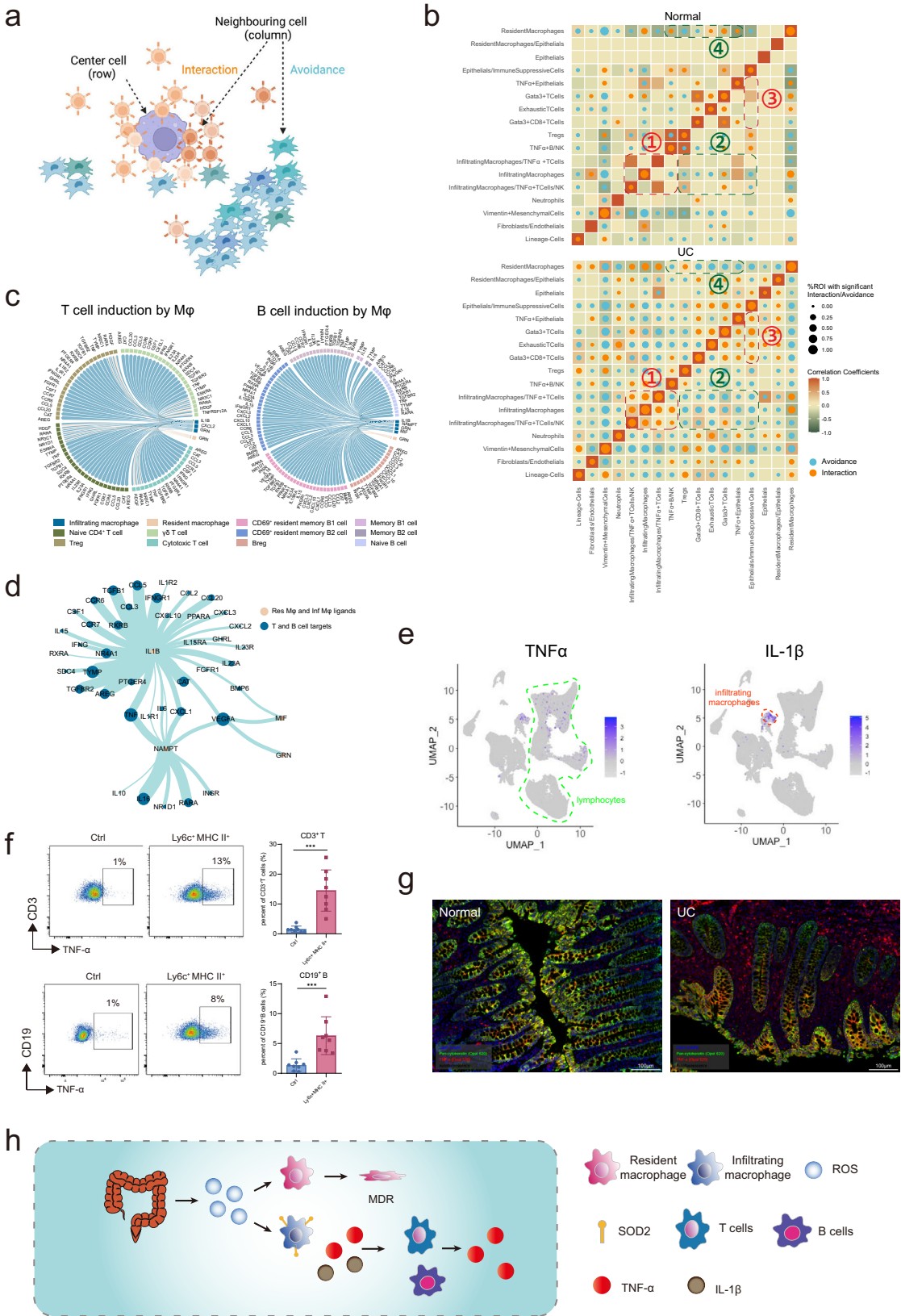

Following another three PBS-TB washes, sections were incubated with an antibody cocktail at 4 °C overnight. The antibodies were mostly labeled with metals using the Maxpar® X8 Antibody Labeling Kit (Fluidigm). Briefly, an X8 polymer was linked to the reduced antibody, and a specific metal was loaded onto the polymer. The final product panel is presented in Supplementary Table S1. After antibody

incubation overnight at 4 °C, three additional PBS-TB washes were performed. The sections were incubated with a 1.25 μM Intercalator-Ir solution (201192B, Fluidigm) in PBS-TB for 30 min at RT to label nuclei, followed by two PBS-TB washes and one final ddH$_2$O wash. An imaging mass cytometer (Fluidigm, Hyperion) was employed to scan the prepared sections to generate the multiplexed images.

**Fig. 7 | Infiltrating macrophages induces a spatial shift of TNF-α production.**
**a** Schematic of spatial cell-cell cross talk analysis. Cell-cell crosstalk was categorized into interaction and avoidance co-occurrence patterns. **b** Circles indicating patterns of cell-cell interactions/avoidance for normal and UC (Normal, n = 19; UC, n = 33 from two independent experiments). The circle size shows the percentage of ROIs with significant interaction/avoidance determined by the permutation test. Rows represent the cell type of interest (center cell) and columns represent other cell types surrounding the interest cell type (neighboring cell). Colors in the heatmap indicate the Pearson correlation between cell types across all ROIs in normal and UC respectively. **c** Circos plots showing cell-cell communications from inflammatory/resident macrophages to T/B cells involving chemokines and cytokines. **d** Network graph summarizing cell-cell communications between macrophages and T/B cells. Vertex sizes and edge widths scale to numbers of interacted cell types. **e** UMAP identifying distinguishable expression of TNF-α in lymphocytes and IL-1β in infiltrating macrophages, T cells, and B cells. **f** Ly6c⁺ MHC II⁺ inflammatory macrophages were sorted from the colon in DSS treated mice and cultured with lymphocytes from mesenteric lymph nodes. Representative plots and statistics results (ctrl, n = 8; Ly6c⁺ MHC II⁺, n = 8 from two independent experiments) were shown for TNF-α production from CD3⁺ T cells and CD19⁺ B cells after the co-cultivation as mean +/− SD and performed with two-tailed T test (***p < 0.001). Exact P values were provided in the Source Data file. **g** mIHC staining of TNF-α production. Intestinal tissues from normal and UC patients were stained with Pan-cytokeratin (green), TNF-α (red), and DAPI for DNA (blue). All markers were listed in the legend on the image. Scale bars, 100 μm. Representative plot from 5 samples (2 independent experiments). **h** The abstract diagram interprets the mechanism of the resident macrophage disappearance effect. ROS was elevated as the first defense against the invading bacteria. At the same time, immune-suppressive resident macrophage was wiped out purposely due to their sensitivity to ROS and replaced by inflammatory macrophage which was resistant to ROS based on the high level of SOD2. Furthermore, inflammatory macrophages played a key role in forming the inflammatory cellular network by producing TNF-α and IL-1β. Resident macrophages might go through cell death due to high ROS stress.

## IMC image preprocessing and cell segmentation pipeline

The IMC data analysis pipeline can be divided into four steps, spillover signal compensation, image denoise, image contrast enhancement, and cell segmentation. For the spillover signal in each channel, we use the spillover matrix defined previously[28]. In the image denoise step, we applied median filtering for noise suppression[29]. We set the window size of the median filter as 3*3, where each output pixel contains the median value in the 3-by-3 neighborhood around the corresponding pixel in the input image. To enhance the contrast of each image, we use the Matlab function *imadjust* for intensity adjustment, by which the intensities can be better distributed utilizing the full range of intensities i.e., 0–255. To segment individual cells or components in different channels of IMC images, we applied one connectivity-aware segmentation method described previously[31]. We applied the Matlab function *regionprops* to detect connected components in the image for cell segmentation. As to other membrane channels, we remove the artifacts if their corresponding distance to the nearest nuclei centroids are larger than 15 pixels. All the codes are implemented by matlabR2017a that are publicly available at https://github.com/shaoweinuaa/NC2022_Ulcerative_colitis.

## IMC downstream analysis pipeline

Marker expression was firstly range normalized to the 99th percentile across all cells for each channel separately and R package Harmony (version 0.1.0) was applied to align batch effects. Then we used Rphenograph (version 0.99.1), an R implementation of Phenograph[66], and 100 nearest neighbors to cluster cells and cluster means were displayed as heatmap and used for annotation. The cellular neighborhood for each cell was captured using windows consisting of the 20 nearest neighboring cells as measured by Euclidean distance between X/Y coordinates. These windows were then clustered by their compositions concerning the 17 cell types using K-means clustering. With k = 15, each cell was then allocated to the neighborhood (CN) that its surrounding window was. CN assignment was validated by overlaying their Voronoi allocation graphs on the original tissue IMC images.

## Spatial analysis

To investigate cell-cell interactions, a permutation test method test interactions() implemented in imcRtools (version 1.0.2)[67] was used to determine whether interactions/avoidances between each cell type within each CN occurred more frequently than random observations. The tensor of CN and cell type distributions for each patient was produced by computing the frequency of each cell type in each CN for each patient. This tensor was then split into Normal and UC along the patient direction. Non-negative Tucker decomposition as implemented in the Tensorly (version 0.7.0) Python (version 3.7.10) package was applied to each tensor[68]. The rank of 4 was selected by assessing the decomposition loss. All the codes are implemented are public available at https://github.com/shaoweinuaa/NC2022_Ulcerative_colitis.

## Fluorescence-activated cell sorting (FACS) and Cy-TOF data acquisition

Tissue obtained by biopsies or mice colon were digested in a culture medium supplemented with 0.6 mg/mL collagenase IV (17104019, Gibco) and 0.01 mg/mL DNase I (11284932001, Merck) in a 170-rpm constant temperature shaker (Eppendorf) at 37 °C for 1 h. Digested tissues were passed through a 70 μm cell strainer (352340, BD) to acquire single-cell suspensions before centrifuging at 300 g for 5 min and resuspending in 35% Percoll (P4937, Sigma). After another centrifugation at 500 g for 5 min, cells were resuspended in 10 mL of blood lysis buffer (555899, BD) for 10 min at RT to eliminate red blood cells. Cell suspensions were centrifuged at 300 g for 5 min and then incubated with 2.5 μg/mL Fc blocker (156604 for mouse; 422302 for human, BioLegend) on ice for 15 min. The suspensions were further incubated with fluorochrome- or metal-labeled antibodies at 4 °C for 30 min. The antibody panel is listed in Supplementary Table S1. The samples were washed and resuspended in PBS and supplemented with 2% FBS. For intracellular staining, BD kits (555028) were used. For FACS analysis, a five-laser flow cytometer (BD Bioscience, Fortessa) was applied. For Cy-TOF analysis, Helios (Fludigm) was applied. The FACS data were analyzed with FlowJo software (TreeStar).

To monitor the capacity of Ly6c⁺ MHC II⁺ infiltrating macrophages stimulating in vitro TNF-α production of T cell and B cell, CD45⁺ CD11b⁺ F4/80⁺ Ly6c⁺ MHC II⁺ infiltrating macrophages were isolated and purified by cell sorting from inflamed colons in DSS models. T cells and B cells were isolated and purified from mesenteric lymph nodes of mice from the same model. Infiltrating macrophages and T/B cells were mixed at a 1:20 ratio. Flow cytometry cell analysis was performed after co-culture for 6 h and blocking by Golgi inhibitor for 4 h (555029, BD).

## CyTOF data analysis

CyTOF data was pre-processed with FlowJo to manually gate live cell population from Flow cytometry standard (FCS) files. The resulting populations were fed to Cytosplore software[69] for major cluster identification. Spanning-tree Progression Analysis of Density-normalized Events (SPADE) algorithm was applied on hyperbolic arcsinh the transformed data with a cofactor of 5, for clustering. t-distributed Stochastic Neighbor Embedding (tSNE) was performed for dimensionality reduction. Visualization was implemented in ggplot2.

## Single-cell RNA-seq library construction

Four healthy control subjects and 4 UC patients were enrolled at the Health Examination Center of Beijing Chaoyang Hospital of Capital Medical University. Written informed consent was obtained and ethical approval was granted by the ethics committee of the Beijing Chaoyang Hospital, Capital Medical University. 4 pinch biopsy specimens from the healthy volunteers served as healthy control (HC group). For each of the 4 patients, 1 pinch biopsy specimen was

collected from the inflamed colon region as the UC group, and 1 pinch biopsy specimen from the normal ascending colon of patients served as self-control (UCSC group). All the biopsy specimens were taken by endoscopic procedure by professional surgeons listed in the author list.

Biopsy specimens were collected into RPMI 1640 medium on ice and processed immediately into single cells. Single cells were dissociated from the intestinal samples obtained by biopsies through the method mentioned in the FACS analysis. Briefly, the single cells were loaded into Chromium microfluidic chips with v3 chemistry, then barcoded with a 10× Chromium Controller (10X Genomics, Pleasanton, CA, USA). RNA from the barcoded cells was subsequently reverse-transcribed and sequencing libraries were constructed with reagents from a Chromium Single Cell v3 reagent kit (10X Genomics) according to the manufacturer's instructions. Single-cell RNA-seq library construction was performed by OE Biotech Co, Ltd (Shanghai, China).

Library sequencing was performed at Novogene Co., Ltd (Tianjin Novogene Technology Co., Tianjin, China) with Illumina HiSeq 2000 according to the manufacturer's instructions (Illumina, San Diego, USA).

### Single-cell RNA-seq analysis

For the droplet-based 10× Genomics data, the Cell Ranger Single-Cell toolkit provided by 10× Genomics was applied to align reads and generate the unique molecular identifier (UMI) matrix for each sample with Grch38 as the reference genome. Seurat (version 4.1.0) was used for downstream analysis[70]. Cells with fewer than 200 and above 6000 detected genes were filtered out, as well as cells with a high proportion of mitochondrial gene counts per cell (>25%). Batch effects across different individuals were removed by firstly identifying anchors using the FindIntegrationAnchors() function, followed by IntegrateData() to obtain a batch-corrected space. For visualization, the dimensionality was reduced using UMAP implemented by RunUMAP() function. Major Clusters were identified by a shared nearest neighbor (SNN) modularity optimization-based clustering algorithm using FindNeighbors() and FindClusters() with default settings. Cells of each major immune cell type including Macrophage, T, and B cells were extracted from the raw matrix and further clustering was performed to obtain the subpopulation structures. FindMarkers() was applied to identify cluster-specific higher expressed genes. These genes were used for cluster annotation together with known lineage-specific markers.

**Differential expression and functional enrichment analysis.** Differential expression analysis between groups was performed using FindMarkers() function provided with Seurat package (version 4.1.0). The cut-off value was set at adjusted $p$-value < 0.05 and |Fold-Change| > 1.5 for identifying upregulated and downregulated genes. Hallmark gene set enrichment analysis was carried out using R package clusterProfiler (version 4.2.2)[71], with a significant enrichment threshold of BH-corrected $P$-value < 0.05.

**Regulation network and intercellular communication analysis.** SCENIC is an algorithm that can reconstruct gene-regulatory networks and identify stable cell states from single-cell RNA-seq data. The regulon matrix of inflammatory and resident macrophages was generated by SCENIC (version 1.2.4) using the corresponding count matrix. In brief, SCENIC first identifies potential TF targets based on coexpressions, performs the TF-motif enrichment analysis, and identifies the direct regulons. SCENIC then scored all cells for the activity of each regulon by calculating the enrichment of the regulon as an area under the recovery curve (AUC)[50]. Intercellular communications between subtypes of macrophages, T, and B cells were evaluated using NicheNet[56]. All the codes are implemented are public available at http://ibrain.nuaa.edu.cn/_upload/tpl/02/be/702/template702/code.zip.

### Statistical analysis

No statistical method was used to predetermine sample size. No data were excluded from the analyses. The experiments were not randomized. Raw data obtained from FACS and single-cell analysis were copied into GraphPad software. Statistical tests were selected based on the appropriate assumptions for the data distribution and variability characteristics. Sample data were analyzed by a two-sided Wilcoxon rank-sum tests to identify statistically significant differences between the two groups. One-way ANOVA with the Bonferroni post-test was used to identify differences among three or more groups. The data are represented as the means ± SEMs. A $p$-value < 0.05 indicated statistical significance.

### Reporting summary

Further information on research design is available in the Nature Portfolio Reporting Summary linked to this article.

## Data availability

The data underlying this article are available at: https://www.ncbi.nlm.nih.gov/geo/query/acc.cgi?acc=GSE231993. The IMC raw files, pre-processed files, exported matrix files are deposited into OMIX databases: OMIX001059 (Public available at https://ngdc.cncb.ac.cn/omix/release/OMIX001059). Source data are provided with this paper.

## Code availability

All the codes related to the analysis are publicly available at https://github.com/shaoweinuaa/NC2022_Ulcerative_colitis.

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

## Acknowledgements

This work was supported by the National Natural Science Foundation of China (grant 82000489 to J.D., 82173078 to J.S., grant 82070559 to X.Liu, grant 61902183, 62272226 to W.S., 62136004 to D.Z., 82270565 to X.C., grant 8218810 to J.S.). National Key Research and Development Program of China (grant 2019YFA0803000 to J.S., and grant 2019YFA0905600 to H.H.), the Excellent Youth Foundation of Zhejiang Province (R22H1610037 to J.S.), the Zhejiang Provincial Natural Science Foundation (grant 2022C03037 to J.S., and Y19H030059 to J.D.). And we are grateful to the participating patients for their contribution to our research. We also thank OE Biotech Co, Ltd (Shanghai, China), Novogene Co., Ltd (Tianjin Novo-gene Technology Co., Tianjin, China), and MobiDrop (Zhejiang) Co., Ltd. for the technical and analytical support for the single-cell analysis.

## Author contributions

Conceptualization: J.S.; Methodology: J.D., J.S., JunleiZ., and J.S.; scRNA-seq sample preparation: J.D., G.L., X.Liu, and X.C.; IMC panel: J.S., JunleiZ., and X.W.; mIHC: X.W.; Patient sample isolation: J.D., J.Lu, M.N., JieZ., G.L., X.Liu, and X.C.; Hyperion operation: Y.Z.; Pathological support: F.C. and J.Li; Center core facility management and assistance: T.F. and X.Liang; IMC image processing: Q.Z., D.Z., and W.S.; Data ana-lysis: J.S., JunleiZ., Bioinformatic analysis: L.W., Y.Z., and H.P.; Writing: J.S., and JunleiZ.; Editing: J.S.; Supervision: J.S.; Funding acquisition: J.S., W.S., X.Liu, H.H., and J.D.

## Competing interests

The authors declare no competing interests.

## Additional information

[1]Department of Gastroenterology, the First Affiliated Hospital, Zhejiang University School of Medicine, Hangzhou 310002, China. [2]Department of Hepato-biliary and Pancreatic Surgery, the First Affiliated Hospital, Zhejiang University School of Medicine, Hangzhou 310002, China. [3]Zhejiang Provincial Key Laboratory of Pancreatic Disease, the First Affiliated Hospital, Zhejiang University School of Medicine, Hangzhou 310002, China. [4]Zhejiang University Cancer Centre, Zhejiang University, Hangzhou 310002, China. [5]Department of Clinical Pharmacy, the First Affiliated Hospital, Zhejiang University School of Medicine, Hangzhou 310002, China. [6]Central Laboratory, First Affiliated Hospital, School of Medicine, Zhejiang University, Hangzhou 310002, China. [7]Pathology

Department, The First Affiliated Hospital, Zhejiang University School of Medicine, Hangzhou 310002, China. [8]College of Computer Science and Technology, Nanjing University of Aeronautics and Astronautics, No. 29 JiangJun Road, Jiang Ning District, Nanjing, Jiangsu 211106, China. [9]MobiDrop (Zhejiang), No. 455 Heshun Road, Tongxiang, Zhejiang 314500, China. [10]Department of Gastroenterology, Beijing Chaoyang Hospital, Capital Medical University, Chaoyang District, Beijing 100024, China. [11]Frontiers Science Center for Synthetic Biology, School of Chemical Engineering and Technology, Tianjin University, Tianjin 300072, China. [12]State Key Laboratory of Experimental Hematology, National Clinical Research Center for Blood Diseases, Haihe Laboratory of Cell Ecosystem, Institute of Hematology & Blood Diseases Hospital, Chinese Academy of Medical Sciences & Peking Union Medical College, Tianjin 300020, China. [13]Department of Hepato-Gastroenterology, Tianjin Medical University General Hospital, Tianjin Medical University, Tianjin 300000, China. [14]These authors contributed equally: Juan Du, Junlei Zhang, Lin Wang, Xun Wang. ✉e-mail: dujuan@zju.edu.cn; caoxiaocang@ihcams.ac.cn; liuxinjuan@mail.ccmu.edu.cn; shaowei20022005@nuaa.edu.cn; shengjp@zju.edu.cn

