## [Peer Review File · Nature Communications]

Selective oxidative protection leads to tissue topological changes orchestrated by macrophage during ulcerative colitisREVIEWER COMMENTS

Reviewer #1 (Remarks to the Author):

Du et al. present a multi-omic study of ulcerative colitis (UC), initially profiling a small cohort of patient samples spatially with imaging mass cytometry and following up with some single cell RNA seq and mouse DSS colitis model in an effort to provide some more mechanistic / dynamic view of the disease. Their main conclusions are that their data support the MDR hypothesis where homeostatic, tissue resident macrophages disappear during the disease process. They claim this is driven by increase ROS (predicted by increased SOD RNA in human samples) and ROS increase and accompanied by apoptosis in the mouse DSS model. While this overall set of conclusions seems consistent with what is expected in the UC literature, the authors analysis only actually supports increased immune infiltrations (see below). Furthermore, the mechanism (figure 7) proposed is a qualitative interpretation of independent observations in unreated samples, systems, and analyses. Moreover, there are numerous technical artifacts in the data presented in the manuscript which undercut the strength and validity of the underlying conclusions. Major concerns are as follows:

-IMC data accuracy / quality – Inspection of the healthy colon imaging data presented in figure 1 reveal numerous imaging / staining artifacts that would not be biologically anticipated and unfortunately seem to have been extracted and carried through the subsequent analyses presented in the paper (subsequent heatmaps and clustering data). These include the co-expression of (true-positive) cytokeratin staining in the glandular region with numerous immune cell markers including CD3, CD11b, CD20, CD11c, CD45R0/CD45RA..ect. TNFa inflammatory Cytokine expression in a curious gradient pattern only in the epithelial cells of healthy, non-inflamed tissue. Markers like CD3 don't broadly overlap with CD4 or 8 (as expected), CD45RA/RO, expected to have mutually exclusive staining, broadly overlap. B7-H3 seems to be coming primarily from bare slide regions. Many other markers measured (most cytokine and checkpoint probes) do not have any believable staining patterns in any images provided. Given that all of these are 'home made' IHC reagents it seems that the authors could stand to share some basic validation and titration data on tissue controls to establish specificity. Moreover, it appears some of these artifacts might be due to isotopic and oxidation interference from abundant markers (which can possible be compensated for PMC5981006) – but the artifacts seem too numerous to fix, let alone mention here.

- MDR or immune infiltration – all the data supporting the MDR hypothesis here deals with increased in frequency of inflammatory cells (and thus decrease in tissue resident macs) – but changes in frequency don't mean disappearance – it could just be an increase in other cells. The authors need to go back to the images and actually see if the absolute #s per unit area of the non-inflammatory, tissue resident macs have actually changed versus healthy.

- ROS / Cytokines / Apoptosis – While the authors try to thematically connect these as a mechanism – they aren't actually connected in the sample samples / system / experiment – even though they could derive much of these observations directly from the IMC / human tissue data in situ – they don't. Combined with an odd Western blot demonstration of caspase 3 increase (in what cells?) – instead of directly by cytometry or imaging as described in the methods and tables – the authors really fail to observe MDR directly in human UC samples let alone show its relationship to cytokine production of ROS.

- Data sharing – the possible analysis on the imaging and scRNA-seq data is far from comprehensive and the authors make no effort to share data negating any data resource aspects of the manuscript. This should include raw imaging and cell files, image segmentation masks and cluster assignments.

Reviewer #2 (Remarks to the Author):

The authors have used imaging mass cytometry, flow cytometry and single cell RNA sequencing to profile the cellular landscape in Ulcerative colitis. The technologies provide complementary perspectives of the UC ecosystem and appear to have validated the authors' preconceived hypotheses as well as provide evidence of undescribed cellular relationships. This study is well designed and should provide a valuable contribution to the field.

The current description of the analytical methods is poor, making it difficult to 1) assess the robustness of the results and 2) ensure the results are reproducible. The authors should substantially improve the methods section regarding the analysis and processing of both the scRNA-seq and IMC data so that this can be assessed. This should include moving or duplicating the approaches described in the results to the methods section.

The authors have made their raw scRNAseq data publicly available. I was not able to find the IMC data. This should also be shared.

The authors should share their processed data including cell type labels and x-y coordinates.

All code for performing the scRNAseq and IMC analysis should also be publicly shared to ensure reproducibility of the results and to enable assessment of the robustness of the results.

Analysis that is not coded should be described in detail. For instance, "we used CellProfiler to acquire mask files" is not informative at all.

The authors should include version numbers when describing software as version changes can alter results and interpretation.

The authors describe significant difficulties segmenting dense regions of lymphocytes. This is not surprising. However, they attribute this to the resolution of the IMC as opposed to a membrane marker not being used for segmentation. Could the authors provide figures to demonstrate this is a technological limitation as opposed to an analytical one.

It is not clear what the purpose of the cellular neighbourhood analysis is. While the authors describe changes in the proportion of cellular neighbourhoods between disease severities, they attribute each of these cellular neighbourhoods to single cell types. It seems an opportunity to describe interactions between multiple cell types has been missed here.

The observations made regarding TNFa appear exciting. The authors have used NicheNet to infer cell-cell communication from the scRNAseq. It would be informative to see if these results are consistent with communication implied by physical distances in the IMC.

Flow cytometry, scRNAseq and IMC are all complementary technologies that provide different perspectives of the cellular composition of a tissue and suffer different technological limitations. To assess these differences, it would be useful to demonstrate which cell types had similar or different frequencies across the technologies.

The manuscript does not appear to reference other work using other cellular assays, such as CyTOF, to profile inflammatory bowel diseases, such as the one published by this journal in 2019. DOI: 10.1038/s41467-019-10387-7 . It would have been insightful to see if the results are consistent with both the scRNAseq and IMC.

While only just published, it might be worth commenting on the recent manuscript in Gastroenterology that also uses IMC to profile inflammatory bowel disease. "Highly Multiplexed Image Analysis of Intestinal Tissue Sections in Patients With Inflammatory Bowel Disease."

The manuscript would benefit from a further round of edits for grammar. There are many inconsistencies in tense and plurals as well as poorly structured sentences such as Line 121: "However, if MDR happens in UC and whether same coagulation process occurred during UC progression was yet not understood."

Reviewer #3 (Remarks to the Author):

The manuscript deals with an interesting topic concerning the compositional and spatial changes in the intestinal ecosystem in ulcerative colitis (UC). The study provides a better understanding of the topological changes that occur in UC, using a methodology that included transcriptomics and proteomics.

Therefore, I would like to make the following comments and suggestions:

1. The study used both human samples and animal models. The Abstract does not specify what kind of samples were associated with each result's description. The same applies to the end of the "Introduction" section. However, I myself would prefer not to include results in this section as the authors did.
2. In the "Material and Methods" section, there is a lack of information on the methods employed for the human samples. For example, there is no information on what kind of samples were used for the single cell RNAseq analysis.
3. Concerning the characterization of the human samples, I suggest that the authors include more clinical information about the patients and the healthy controls' cohort. I suggest to include a table with detailed demographic and clinical data with the median (maximum and minimum) of the numerical variables and the absolute number of the categorical variables.
4. Concerning the DSS colitis, the authors could have used the induced colitis score that assesses, in addition to weight loss, the presence of occult blood in the feces, and feed intake.
5. The authors stated on page 6/line 150 that the colon samples came from surgical resections or biopsies. What about the healthy donors? What were the reasons for surgical resections or biopsies? My observation on this fact would be that, in the result section, these tissues were labeled for different cell types, which were used for comparison between healthy controls and UC patients.
6. On page 17, lines 414-420, a co-culture experiment involving mouse Ly6c+MCH II macrophages is described. It would be interesting to add this methodology to the "Materials and Methods" section.
7. In the Results section Line 307-312: How was it possible to determine the exact day for this analysis if the sacrifice and colon sample collection was done 5 weeks after the DSS treatment?
8. The legend of Figure 1C lacks information on what the arrows are signaling.
9. I suggest the author include a descriptive scheme of the clusters in the Supplementary Figure S2, to facilitate the interpretation of the data. Besides, I suggest them to increase the size of the color scales to make them readable and to know the expression level in each graph (Plot).
10. In the "Discussion" section, it would be interesting to provide more detailed comparison between the data findings and what is already present in the literature. Even if the data is controversial, or even complementary, I believe this section of the manuscript should be developed more.
11. The phrase (Line 433-436) is unclear. I suggest that it be rewritten.

12. In the "Abbreviations" section, it is necessary to add the abbreviations: FACS, CN, DEG, PMA, and LPS. In addition, for the acronyms PMA and LPS (page 15, lines 372 and 373 respectively) the meaning of each acronym was missing. The same was true of CN, on page 16, subsection "Cellular neighborhood changes during UC".

13. Please check the Reference formatting guideline, which states, "Authors should be listed surname first, followed by a comma and initials of given names." Please revise the authors' names in the "References" section.

Overall, it is a well written article. The study addresses an interesting aspect of ulcerative colitis, but needs revision.

Dear reviewers,

Many thanks for all your careful reading and advice. We are grateful for the constructive feedback provided by all the reviewers, and we agree with most of the criticisms and suggestions put forward, which helped us to improve our manuscript in many different aspects. Please see our point-to-point reply below.

Reviewer #1 (Remarks to the Author):

Du et al. present a multi-omic study of ulcerative colitis (UC), initially profiling a small cohort of patient samples spatially with imaging mass cytometry and following up with some single cell RNA seq and mouse DSS colitis model in an effort to provide some more mechanistic / dynamic view of the disease. Their main conclusions are that their data support the MDR hypothesis where homeostatic, tissue resident macrophages disappear during the disease process. They claim this is driven by increase ROS (predicted by increased SOD RNA in human samples) and ROS increase and accompanied by apoptosis in the mouse DSS model.

1. While this overall set of conclusions seems consistent with what is expected in the UC literature, the authors analysis only actually supports increased immune infiltrations (see below). Numerous technical artifacts in the data presented in the manuscript which undercut the strength and validity of the underlying conclusions. Major concerns are as follows:

Reply: dear reviewer, thanks for the critical and constructive comments. We totally understand that the biggest concern is that MDR is only a side effect observed due to the heavy infiltration of immune cells, especially myeloid cells. However, we are quite confident that MDR in UC is a true phenomenon based on several reasons below.

Please note macrophage disappearance reaction (MDR) was first reported in the peritoneum in response to certain stimuli ¹ more than two decades ago. During our continued study focusing on the macrophage, we found that MDR is a general phenomenon in many infection and inflammation conditions. We and our collaborators could observe MDR in the lung, kidney, liver, and colon in response to virus, fungi, malaria, and chemical-induced colitis ²⁻⁶. Thus, we are quite sure that MDR is a true and general phenomenon in response to different stimuli but not due to the side effect of heavy infiltration of immune cells. Large-scale scRNA-seq analysis of COVID-19 patients' samples also illustrated macrophage type switch ⁷.

And we believed that MDR is not so well characterized, partially due to the complexity of macrophage subsets. Most tissues are comprised of both resident and infiltrating macrophages and it is important to note the MDR occurred only for the tissue-resident macrophage across tissues, such as the alveolar macrophage in the lung, F4/80^{hi} resident macrophage in the kidney, Kupffer cell in liver and CX3CR1^{hi} resident macrophage in colon⁸. And it is critical to distinguish resident and infiltrating macrophages well, before a detailed study of MDR. We agree that the novelty of current manuscript is not the infiltrating of immune cells. We want to emphasize macrophage subpopulation switch due to the resident macrophage specific MDR, and the mechanism, consequence of such macrophage switch.

Claiming MDR in UC is only due to heavy infiltration of immune cells is not accurate and we believe to separate the resident and infiltrating macrophages well for the proper study of MDR in UC, which is more challenging in intestinal space than other tissues due to two major reasons. First, it is easier to lose the resident macrophage during the vigorous isolation process of intestinal tissue, especially when the mucus layer is removed. Second, intestinal resident and infiltrating macrophages are quite similar. For example, both of them are F4/80⁺ and CX₃CR1⁺ in the mouse. And the resident intestinal macrophage in mice is F4/80^{hi} CX₃CR1^{hi}, while mouse intestinal infiltrating macrophage is F4/80^{int} CX₃CR1^{int}^{6,8}. Thus, it is not surprising that the MDR phenomenon is more difficult to observe in intestinal tissues. And MDR for resident macrophage in UC is not only observed by us but also by other experts like Steffen Jung in the macrophage field⁶.

In our previous and current study of the intestinal system, we defined the resident macrophage as F4/80^{hi} CD169⁺ in mice^{8,9} and CD68⁺ CD169⁺ CD11b^{low} in humans (current manuscript). Such definition could delineate resident and infiltrating macrophages well in both FACS and IMC experiments after the proper cell dissociation step, which ensured the accurate interpretation of our results.

2. Furthermore, the mechanism (figure 7) proposed is a qualitative interpretation of independent observations in unrelated samples, systems, and analyses.

Reply: dear reviewer, thanks for the constructive criticism and we agree that our previous experiment settings were unrelated. We have upgraded our experiments, figures, and texts. For example, we removed the unrelated and artificial *in vitro* LPS challenge experiment and updated the ROS and macrophage staining based on the patients' samples. Please refer to main Figure 5 and the results section on page#12 line#287 to page#13 line#312.

3. Moreover, there are numerous technical artifacts in the data presented in the manuscript which undercut the strength and validity of the underlying conclusions.

-IMC data accuracy / quality – Inspection of the healthy colon imaging data presented in figure 1 reveal numerous imaging / staining artifacts that would not be biologically anticipated and unfortunately seem to have been extracted and carried through the subsequent analyses presented in the paper (subsequent heatmaps and clustering data).

Reply: thanks for the critical comments provided. The imaging/staining artifacts were mainly due to the resolution limit (1 μ m) of IMC technology, resulting in overlapped marker detection. Please note such limitations existed in other labs relying on IMC as well. For example, in the Nature paper “The single-cell pathology landscape of breast cancer” from Bernd Bodenmiller’s group, T and B cell markers were located together in one cluster. In addition, we have verified all our suspicious IMC staining by mIHC (resolution at 220 nm) and updated the results from page#7 line#151 to line#166. Please also refer to Figure S4 on page for detailed pictures.

4. These include the co-expression of (true-positive) cytokeratin staining in the glandular region with numerous immune cell markers including CD3, CD11b, CD20, CD11c, CD45R0/CD45RA..ect.

Reply: IMC experiment validation was updated, and we also verified the same staining via mIHC staining (Figure S1 to 4). Please refer to results from page#7 line#151 to page#7 line#166 and Figure S4 for a detailed description. Please refer to Figure S1-3 for detailed validation results of the IMC experiment. As explained above, the marker overlapping was mainly due to the resolution limit of IMC, but the staining was true. We think it’s better defined as co-staining rather than co-expression.

We agreed that such an artifact was carried through the subsequent analysis. In fact, such an artifact can’t be improved by increasing resolution limit. Marker overlapping also exists in mIHC imaging with a much higher resolution limit (Figure S4). However, such overlapping might just indicate the close interaction of neighbor cells and won’t hamper the conclusion drawn in our manuscript.

5. TNFa inflammatory Cytokine expression in a curious gradient pattern only in the epithelial cells of healthy, non-inflamed tissue.

Reply: thank the reviewer for the reminder and TNF- α staining was also repeated by IMC and mIHC (Figure S3 and main Figure 7). And we totally agreed that the TNF- α staining pattern is very intriguing. TNF- α is mainly detected in the tropical epithelial region in a healthy colon while it is mainly detected in the lymphocytes within the LP during UC. And the spatial shift of TNF- α production was one of the major sections of the results in the updated manuscript. Please refer to main Figure 7 and results from page#14 line#343 to page#16 line#375.

6. Markers like CD3 don't broadly overlap with CD4 or 8 (as expected).

Reply: please note, CD3 in activated T cell is reduced. We performed CD3, CD4, and CD8 staining in healthy control and UC region, and it is clear to see CD3 staining in CD4⁺ and CD8⁺ T cells could be separated (Figure S4) due to lowered expression level after T cell activation. Please refer to page#7 lines #159 to 163 and Figure S4 for a detailed description.

7. CD45RA/RO, expected to have mutually exclusive staining, broadly overlap. B7-H3 seems to be coming primarily from bare slide regions.

Reply: we noticed CD45RA/RO co-staining pattern in our previous IMC publication on HCC already ¹⁰. In fact, we felt that CD45RA/RO staining is quite overlapped. Also, in our Cy-TOF analysis and mIHC staining, we could observe co-staining of CD45RA/RO (Figure S4). Thus, we felt that CD45RA/RO may not be so mutually exclusive. For detailed results and figure, please refer Figure S4 and the results section on page#7 lines #163 to 166. We have verified each of the antibodies used. For B7-H4 validation results, please refer to Figure S1 for detail staining.

8. Many other markers measured (most cytokine and checkpoint probes) do not have any believable staining patterns in any images provided.

Given that all of these are 'home made' IHC reagents it seems that the authors could stand to share some basic validation and titration data on tissue controls to establish specificity. Moreover, it appears some of these artifacts might be due to isotopic and oxidation interference from abundant markers (which can possibly be compensated for PMC5981006) – but the artifacts seem too numerous to fix, let alone mention here.

Reply: thanks for the constructive criticism. As mentioned above, most “artifacts” mentioned here were due to the resolution limit of IMC technology and the current analysis pipeline relying on cell segmentation can’t separate two very close cells.

Thanks for the suggestion. To increase the IMC multiplexity to 40, we have to utilize many home made IHC antibodies and all our antibody-metal combinations were based on the previous work and our collaborators’ validated panel. The antibody validation results for intestinal tissue were included in the updated Figure S1-3. Please refer to Figure S1-3 and the results section on page#6 lines #129-131.

Thanks for the paper suggestion and we embedded their compensation method in our newly developed analysis pipeline. Please refer to the results section on page#7 lines#167-173 and the methods section on page#20 lines #481-495.

9.- MDR or immune infiltration – all the data supporting the MDR hypothesis here deals with increased in frequency of inflammatory cells (and thus decrease in tissue resident macs) – but changes in frequency don’t mean disappearance – it could just be an increase in other cells. The authors need to go back to the images and actually see if the absolute #s per unit area of the non-inflammatory, tissue resident macs have actually changed versus healthy.

Reply: thanks for the critical comment. Absolute number changes of resident macrophages were also included in main Figure 5g to i. Please refer to main Figure 5 and the results section on page#13 lines #305-308 for details. Please also see our reply to your first comment.

10.- ROS / Cytokines / Apoptosis – While the authors try to thematically connect these as a mechanism – they aren’t actually connected in the sample samples / system / experiment – even though they could derive much of these observations directly from the IMC / human tissue data in situ – they don’t. Combined with an odd Western blot demonstration of caspase 3 increase (in what cells?) – instead of directly by cytometry or imaging as described in the methods and tables – the authors really fail to observe MDR directly in human UC samples let alone show its relationship to cytokine production of ROS.

Reply: thanks for the constructive criticism and we totally agree that the WB experiment was artificial and unrelated. We believe the above two questions can be addressed together. We have taken away the WB result and replaced it with ROS and macrophage staining on the

human patient's sample. Please refer to our updated main figure 5e to i. We found ROS level in resident macrophages was high and such high level of ROS level was associated with resident macrophage diminishment and made space for infiltrating macrophages, which induced spatial shift of TNF- α production. Please refer to main Figure 5 to 7 and the results section on page#12 line#287 to page#16 line#375 for details. Please also see our reply to your previous comment#10.

11.- Data sharing – the possible analysis on the imaging and scRNA-seq data is far from comprehensive and the authors make no effort to share data negating any data resource aspects of the manuscript. This should include raw imaging and cell files, image segmentation masks and cluster assignments.

Reply: we are sorry for the incomplete data sharing procedure. scRNA-seq data was deposited. Raw imaging files, pre-processed images, excel files were also deposited. Please refer to our “Data availability” section on page#25 lines #631-636 for details. And cluster assignment information was in Figure S6. Please refer to Figure S6 and the results section on page#8 lines #177 to 189.

Reviewer #2 (Remarks to the Author):

The authors have used imaging mass cytometry, flow cytometry and single cell RNA sequencing to profile the cellular landscape in Ulcerative colitis. The technologies provide complementary perspectives of the UC ecosystem and appear to have validated the authors' preconceived hypotheses as well as provide evidence of undescribed cellular relationships. This study is well designed and should provide a valuable contribution to the field.

The current description of the analytical methods is poor, making it difficult to 1) assess the robustness of the results and 2) ensure the results are reproducible. The authors should substantially improve the methods section regarding the analysis and processing of both the scRNA-seq and IMC data so that this can be assessed. This should include moving or duplicating the approaches described in the results to the methods section.

1.The authors have made their raw scRNAseq data publicly available. I was not able to find the IMC data. This should also be shared.

Reply: we are sorry for the poor methods section, and we described each step in detail in the

updated version. Please refer to the methods sections on page#18 line#420 to page#24 line#609. All the scRNA-seq and IMC raw data and intermediate files were also deposited for public access. Please refer to the “Data availability” section on page#25 lines #631-636. And all the codes used were deposited as well for public access. Please refer to the “Codes availability” section on page#25 lines #637-639 for details.

2.The authors should share their processed data including cell type labels and x-y coordinates.

Reply: we are sorry for the incomplete sharing process. Raw imaging files, pre-processed images, and excel files were deposited for public access. Please refer to the “Data availability” section on page#25 lines #631-636.

3.All code for performing the scRNAseq and IMC analysis should also be publicly shared to ensure reproducibility of the results and to enable assessment of the robustness of the results.

Reply: sorry for the incomplete sharing procedure and all the codes were deposited. Please refer to the “Codes availability” section on page#25 lines #637-639 for details.

4.Analysis that is not coded should be described in detail. For instance, “we used CellProfiler to acquire mask files” is not informative at all. The authors should include version numbers when describing software as version changes can alter results and interpretation.

Reply: sorry for the incomplete sharing procedure and all the analysis parts were explained in detail. We described each step in detail in the updated version of the methods section and version number is included as well. Please refer to the methods sections on page#18 line#420 to page#24 line#609. If there is still anything unclear, we are more than happy to address the questions.

5.The authors describe significant difficulties segmenting dense regions of lymphocytes. This is not surprising. However, they attribute this to the resolution of the IMC as opposed to a membrane marker not being used for segmentation. Could the authors provide figures to demonstrate this is a technological limitation as opposed to an analytical one.

Reply: thanks for the suggestion and we have updated our analysis pipeline. We tried several published analysis pipelines including histoCAT¹¹, MCMICRO¹², and ImaCytE¹³. The pipelines are based on CellProfiler or ImaCytE for segmentation and Ilastik for pixel classification. We managed to adapt their analysis pipelines. However, we think it’s better to

incorporate the pre-processing step. Thus, we decided to update our analysis pipeline as below.

Before single-cell segmentation, the scanned images were subjected to pre-processing to improve image quality. Pre-processing consisted of compensation, denoise, and contrast enhancement steps. Compensation followed principles published in the previous publication¹⁴. In the image denoise step, we applied median filtering for noise suppression¹⁵. To enhance the contrast of each image, we followed the method published previously based on the linear regression model¹⁶. Our pre-processing improved the image quality significantly.

To segment individual cells or components in different channels of IMC images, we applied one connectivity-aware segmentation method described previously^{17,18}. The expression level of markers in each cell was quantified and exported into matrix format. Please refer to main Figure 1 and the results section on page#7 line#167 to page#8 line#189 for details. Methods details were on page#20 line#481 to page#21 line#517.

Please note cell membrane segmentation relied on the continuous connection between the pixels from the same membrane marker in our current protocol. All codes are available and refer to “Codes Availability” on page#25 lines #637 to 639.

6. It is not clear what the purpose of the cellular neighbourhood analysis is. While the authors describe changes in the proportion of cellular neighbourhoods between disease severities, they attribute each of these cellular neighbourhoods to single cell types. It seems an opportunity to describe interactions between multiple cell types has been missed here.

Reply: thanks for the suggestion and as suggested, we performed cell-cell interaction analysis based on CN-CT tensors for normal and UC patients following the pipeline published by Garry P Noland’s group¹⁹. We noticed infiltrating macrophage-centered cell clusters and cellular neighborhoods with TNF- α producing T cells were more enriched in the UC patients. For detailed results, please refer to main Figure 6 and the results section on page#13 line#313 to page#14 line#341.

7. The observations made regarding TNFa appear exciting. The authors have used NicheNet

to infer cell-cell communication from the scRNAseq. It would be informative to see if these results are consistent with communication implied by physical distances in the IMC.

Reply: thanks for the suggestion and we also feel that the TNF- α staining pattern is very intriguing. Physical distance was reflected by the co-occurrence pattern analysis, adapted from Bernd Bodenmiller group²⁰. We noticed infiltrating macrophage interaction with TNF- α production was highly enriched in the UC samples and resident macrophage co-occurrence patterns with neighboring cells were opposite in normal and UC samples. It seemed that infiltrating macrophages caused a spatial shift of TNF- α production in UC regions. Please refer to main Figure 7 and the results section on page#14 line#343 to page#16 line#375 for details. And please also refer to our reply to reviewer#1 point #5.

8. Flow cytometry, scRNAseq and IMC are all complementary technologies that provide different perspectives of the cellular composition of a tissue and suffer different technological limitations. To assess these differences, it would be useful to demonstrate which cell types had similar or different frequencies across the technologies.

Reply: thanks for the suggestion and we have added a parallel comparison of cell frequency in the updated Figure 2 and Figure 4 for IMC, FACS, and scRNA-seq results. The results were consistent with each other across different experimental settings. Please refer to main Figure 2 and 4, and the results section on page#8 line#195 to page#9 line#221, and page#12 line#277-286.

9. The manuscript does not appear to reference other work using other cellular assays, such as CyTOF, to profile inflammatory bowel diseases, such as the one published by this journal in 2019. DOI: 10.1038/s41467-019-10387-7. It would have been insightful to see if the results are consistent with both the scRNAseq and IMC.

Reply: Thanks for sharing the article and a detailed discussion was added. The authors of the mentioned article above also observed CD14⁺ macrophage (likely to be infiltrating macrophage), T cell, and B cell infiltrating in inflamed intestinal regions via Cy-TOF. Please refer to page#16 line#386 to page#17 line#400 for a detailed discussion.

10. While only just published, it might be worth commenting on the recent manuscript in Gastroenterology that also uses IMC to profile inflammatory bowel disease. "Highly

Multiplexed Image Analysis of Intestinal Tissue Sections in Patients With Inflammatory Bowel Disease.”

Reply: Thanks for the reminder and a detailed discussion for the above article was added to the discussion section. Ayano *et al* claimed tissue-resident macrophages and T Cells exhibit increased inflammatory activity in the lamina propria of patients with inflammatory bowel disease based on a 23-marker panel ²¹. We believed the resident macrophage described by Ayano *et al* was in fact infiltrating macrophages, since only CD68 was used in their annotation, and we felt that CD11b⁺ CD68⁺ infiltrating macrophage might dominate the LP region during UC. Please refer to page#16 line#386 to page#17 line#400 for a detailed discussion.

11. The manuscript would benefit from a further round of edits for grammar. There are many inconsistencies in tense and plurals as well as poorly structured sentences such as Line 121: “However, if MDR happens in UC and whether same coagulation process occurred during UC progression was yet not understood.”

Reply: we are sorry for the inconvenience caused during reviewing and thanks for the suggestions. Both analytical part and language were edited by Dr. Vincent Tano and Prof. Sarah Langley, who are both bioinformatical experts and native English speakers. We also required professional academic writing assistance from London insights.

Reviewer #3 (Remarks to the Author):

The manuscript deals with an interesting topic concerning the compositional and spatial changes in the intestinal ecosystem in ulcerative colitis (UC). The study provides a better understanding of the topological changes that occur in UC, using a methodology that included transcriptomics and proteomics.

Therefore, I would like to make the following comments and suggestions:

1. The study used both human samples and animal models. The Abstract does not specify what kind of samples were associated with each result’s description. The same applies to the end of the “Introduction” section. However, I myself would prefer not to include results in this section as the authors did.

Reply: many thanks for the suggestions and we edited the abstract and introduction as suggested and made clear statements about the samples used. The results were mentioned

briefly in the abstract and last paragraph of the Introduction section to give a fast glance at the major conclusion drawn in the current manuscript. Please refer to page#3 lines #52 to 69, and page#4 lines #95 to 100 for details.

2. In the “Material and Methods” section, there is a lack of information on the methods employed for the human samples. For example, there is no information on what kind of samples were used for the single cell RNAseq analysis.

Reply: sorry for the missing information, which was updated in the methods section. Briefly, four healthy volunteers and 5 UC patients were enrolled at the Health Examination Centre of Beijing Chaoyang Hospital of Capital Medical University. Written informed consent was obtained and ethical approval was granted by the ethics committee of the Beijing Chaoyang Hospital, Capital Medical University. 4 pinch biopsy specimens from the healthy volunteers served as healthy control (HC group). For each of the 5 patients, 1 pinch biopsy specimen was collected from the inflamed colon region as the UC group, and 1 pinch biopsy specimen from the normal ascending colon of patients served as self-control (UCSC group). However, data from one of the 5 UC patients were excluded due to scRNA-seq library construction failure. All the biopsy specimens were taken by endoscopic procedure by professional surgeons listed in the author list. Please refer to the details in the Methods section on page#22 lines #548 to 569.

3. Concerning the characterization of the human samples, I suggest that the authors include more clinical information about the patients and the healthy controls’ cohort. I suggest to include a table with detailed demographic and clinical data with the median (maximum and minimum) of the numerical variables and the absolute number of the categorical variables.

Reply: thanks for this suggestion. The table was updated with more clinical information about the patients and the healthy controls’ cohort in main Figure 2a and Table S2.

4. Concerning the DSS colitis, the authors could have used the induced colitis score that assesses, in addition to weight loss, the presence of occult blood in the feces, and feed intake.

Reply : thanks for the suggestion. Results for weight loss and occult blood in feces were added into main Figure 3a. Feed intake data was not recorded in the process.

5. The authors stated on page 6/line 150 that the colon samples came from surgical resections or biopsies. What about the healthy donors? What were the reasons for surgical resections or biopsies? My observation on this fact would be that, in the result section, these tissues were

labeled for different cell types, which were used for comparison between healthy controls and UC patients.

Reply: sorry for the confusing methods section. The samples were taken by the endoscopic procedure. Briefly, four healthy volunteers and 5 UC patients were enrolled at the Health Examination Centre of Beijing Chaoyang Hospital of Capital Medical University. Written informed consent was obtained and ethical approval was granted by the ethics committee of the Beijing Chaoyang Hospital, Capital Medical University. 4 pinch biopsy specimens from the healthy volunteers served as healthy control (HC group). For each of the 5 patients, 1 pinch biopsy specimen was collected from the inflamed colon region as the UC group, and 1 pinch biopsy specimen from the normal ascending colon of patients served as self-control (UCSC group). All the biopsy specimens were taken by endoscopic procedure by professional surgeons listed in the author list. Please refer to the details in the Methods section on page#22 lines #548 to 569.

In main Figure 5, we compared the regulatory network and gene identify across all conditions in Figure 5b. And the DEG analysis was performed for macrophages in UC samples. For detailed results, please refer to main Figure 5 and the results section from page#12 line#287 to page#13 line#312 for details.

6. On page 17, lines 414-420, a co-culture experiment involving mouse Ly6c+MCH II macrophages is described. It would be interesting to add this methodology to the "Materials and Methods" section.

Reply: sorry for the missing part. The methodology detail was added to the "Materials and Methods" section. Please refer to page#22 line#534-540. And the antibody cocktail details were included in Table S2.

7. In the Results section Line 307-312: How was it possible to determine the exact day for this analysis if the sacrifice and colon sample collection was done 5 weeks after the DSS treatment?

Reply: day 1 was the first of the DSS treatment. And 5 mice were sacrificed for each time point, including day 0 (1 day before DSS treatment), day 9, day 14, day 21, day 28, and day 35. Hope we answered the question.

8. The legend of Figure 1C lacks information on what the arrows are signaling.

Reply: thanks for the careful reading and sorry for the missing information. The white arrows and curves point out intestinal glands. And the yellow arrows mark smooth muscles cells labeled with α SMA (yellow) in lamina propria. Missing information was added to the updated figure legend. Please refer to page#29 lines #820 to 823.

9. I suggest the author include a descriptive scheme of the clusters in the Supplementary Figure S2, to facilitate the interpretation of the data. Besides, I suggest them to increase the size of the color scales to make them readable and to know the expression level in each graph (Plot).

Reply: sorry for the inconvenience caused and many thanks for the suggestion. The cell cluster annotation part was updated in Figure S6a. The size of the colour scales and expression level were updated in Figure S5. Please also refer to the results section on page#8 lines #177-189.

10. In the "Discussion" section, it would be interesting to provide more detailed comparison between the data findings and what is already present in the literature. Even if the data is controversial, or even complementary, I believe this section of the manuscript should be developed more.

Reply: thank you for this suggestion. We have discussed several close related papers ²¹⁻²³ and added the corresponding discussion from page#16 line#386 to page#17 line#400. Please also refer to our reply to reviewer #2 comments #9 and 10.

11. The phrase (Line 433-436) is unclear. I suggest that it be rewritten.

Reply: thanks for the suggestion. We have reformed the sentences according to our updated results. And we deleted the "first" claim since a very recent publication scooped us. Please refer to page#16 lines #382-385 for details.

12. In the "Abbreviations" section, it is necessary to add the abbreviations: FACS, CN, DEG, PMA, and LPS. In addition, for the acronyms PMA and LPS (page 15, lines 372 and 373

respectively) the meaning of each acronym was missing. The same was true of CN, on page 16, subsection "Cellular neighborhood changes during UC".

Reply: Thanks for the careful reading and sorry for the missing information. The abbreviations list now is complete in the updated version of our manuscript. The LPS part was deleted since the experiment setting was quite artificial and unrelated. Please refer to the updated "Abbreviations" on page#2 lines #38 to 42, and the results section on page#13 line#313.

13. Please check the Reference formatting guideline, which states, "Authors should be listed surname first, followed by a comma and initials of given names." Please revise the authors' names in the "References" section.

Reply: thanks for the careful reviewing and reference formatting was updated according to the guidelines of Nature Communications.

Overall, it is a well written article. The study addresses an interesting aspect of ulcerative colitis, but needs revision.

Really thank you for your instructive comment.

Reference

- 1 Barth, M. W., Hendrzak, J. A., Melnicoff, M. J. & Morahan, P. S. Review of the macrophage disappearance reaction. *J Leukoc Biol* **57**, 361-367, doi:10.1002/jlb.57.3.361 (1995).
- 2 Teo, Y. J. *et al.* Renal CD169(++) resident macrophages are crucial for protection against acute systemic candidiasis. *Life Sci Alliance* **4**, doi:10.26508/lsa.202000890 (2021).
- 3 Lai, S. M. *et al.* Organ-Specific Fate, Recruitment, and Refilling Dynamics of Tissue-Resident Macrophages during Blood-Stage Malaria. *Cell Rep* **25**, 3099-3109 e3093, doi:10.1016/j.celrep.2018.11.059 (2018).
- 4 Gupta, P. *et al.* Tissue-Resident CD169(+) Macrophages Form a Crucial Front Line against Plasmodium Infection. *Cell Rep* **16**, 1749-1761, doi:10.1016/j.celrep.2016.07.010 (2016).
- 5 Purnama, C. *et al.* Transient ablation of alveolar macrophages leads to massive pathology of influenza infection without affecting cellular adaptive immunity. *Eur J Immunol* **44**, 2003-2012, doi:10.1002/eji.201344359 (2014).
- 6 Zigmond, E. *et al.* Ly6C hi monocytes in the inflamed colon give rise to proinflammatory effector cells and migratory antigen-presenting cells. *Immunity* **37**, 1076-1090, doi:10.1016/j.immuni.2012.08.026 (2012).
- 7 Ziegler, C. G. K. *et al.* Impaired local intrinsic immunity to SARS-CoV-2 infection in severe COVID-19. *Cell* **184**, 4713-4733 e4722, doi:10.1016/j.cell.2021.07.023 (2021).
- 8 Sheng, J., Ruedl, C. & Karjalainen, K. Most Tissue-Resident Macrophages Except Microglia Are Derived from Fetal Hematopoietic Stem Cells. *Immunity* **43**, 382-393, doi:10.1016/j.immuni.2015.07.016 (2015).

- 9 Soncin, I. *et al.* The tumour microenvironment creates a niche for the self-renewal of tumour-promoting macrophages in colon adenoma. *Nat Commun* **9**, 582, doi:10.1038/s41467-018-02834-8 (2018).
- 10 Sheng, J. *et al.* Topological analysis of hepatocellular carcinoma tumour microenvironment based on imaging mass cytometry reveals cellular neighbourhood regulated reversely by macrophages with different ontogeny. *Gut*, gutjnl-2021-324339, doi:10.1136/gutjnl-2021-324339 (2021).
- 11 Schapiro, D. *et al.* histoCAT: analysis of cell phenotypes and interactions in multiplex image cytometry data. *Nat Methods* **14**, 873-876, doi:10.1038/nmeth.4391 (2017).
- 12 Schapiro, D. *et al.* MCMICRO: a scalable, modular image-processing pipeline for multiplexed tissue imaging. *Nat Methods* **19**, 311-315, doi:10.1038/s41592-021-01308-y (2022).
- 13 Somarakis, A., Van Unen, V., Koning, F., Lelieveldt, B. & Holtt, T. ImaCytE: Visual Exploration of Cellular Micro-Environments for Imaging Mass Cytometry Data. *IEEE Trans Vis Comput Graph* **27**, 98-110, doi:10.1109/TVCG.2019.2931299 (2021).
- 14 Chevrier, S. *et al.* Compensation of Signal Spillover in Suspension and Imaging Mass Cytometry. *Cell Syst* **6**, 612-620 e615, doi:10.1016/j.cels.2018.02.010 (2018).
- 15 Lim, J. S. *Two-dimensional signal and image processing*. (Englewood Cliffs, N.J. : Prentice Hall, 1990).
- 16 Nicolas Hautière, J.-P. T., Didier Aubert, Éric Dumont. *BLIND CONTRAST ENHANCEMENT ASSESSMENT BY GRADIENT RATIOING AT VISIBLE EDGES*. Vol. 27 (2008).
- 17 Guadayol, O., Thornton, K. L. & Humphries, S. Cell morphology governs directional control in swimming bacteria. *Sci Rep* **7**, 2061, doi:10.1038/s41598-017-01565-y (2017).
- 18 Nowak, M. R. & Yoonsuck, C. Learning to distinguish cerebral vasculature data from mechanical chatter in India-ink images acquired using knife-edge scanning microscopy. *Annu Int Conf IEEE Eng Med Biol Soc* **2016**, 3989-3992, doi:10.1109/EMBC.2016.7591601 (2016).
- 19 Schurch, C. M. *et al.* Coordinated Cellular Neighborhoods Orchestrate Antitumoral Immunity at the Colorectal Cancer Invasive Front. *Cell* **183**, 838, doi:10.1016/j.cell.2020.10.021 (2020).
- 20 Giesen, C. *et al.* Highly multiplexed imaging of tumor tissues with subcellular resolution by mass cytometry. *Nat Methods* **11**, 417-422, doi:10.1038/nmeth.2869 (2014).
- 21 Kondo, A. *et al.* Highly Multiplexed Image Analysis of Intestinal Tissue Sections in Patients With Inflammatory Bowel Disease. *Gastroenterology* **161**, 1940-1952, doi:10.1053/j.gastro.2021.08.055 (2021).
- 22 Chapuy, L. *et al.* Two distinct colonic CD14(+) subsets characterized by single-cell RNA profiling in Crohn's disease. *Mucosal Immunol* **12**, 703-719, doi:10.1038/s41385-018-0126-0 (2019).
- 23 Rubin, S. J. S. *et al.* Mass cytometry reveals systemic and local immune signatures that distinguish inflammatory bowel diseases. *Nat Commun* **10**, 2686, doi:10.1038/s41467-019-10387-7 (2019).

REVIEWERS' COMMENTS

Reviewer #2 (Remarks to the Author):

Since the previous version the authors have expanded their methods section and provided the code needed to perform the analysis in their manuscript. This has improved the reproducibility of their results. In addition to this they have made their IMC data publicly available which will benefit the broader community. To their credit, the authors have substantially altered the way that they preprocessed their IMC data following feedback from the reviewers. This appears to be concordant with community standards for the analysis of this data type.

Given the improvements in the pre-processing of the IMC data, Figure 1H is very concerning. It is very hard to distinguish how the clusters were annotated as certain cell types given the colours and intensities within the heatmap. For instance, CD4 and CD8 are coexpressed in more than half of the clusters. This is a key figure which should provide confidence in the unsupervised clustering used to define cell types in the experiment. This figure makes it difficult to accept any of the conclusions made that rely on the cell type annotations from the IMC data. Figure S5 reduces this confidence further. Minor points related to Figure 1H, the legend describes the colours as z-scored mean marker expression, it is surprising to see all the values between 0.1 and 0.9. The colours of the cell type frequencies are very difficult to see. It would help if the order of the clusters were consistent between g and h.

Minor points.

T-tests are not appropriate in Figure 2 and Figure 6. I am not sure if this would alter the conclusions substantially though.

Figures were stretched in Figure 3.

Font size in Figure 4 in particular is very small. Font sizes across most panels in most figures are very inconsistent. I am sure it would be beneficial for readers if this was addressed. Maybe the number of panels in each figure need to be reduced to address this.

I appreciate that the authors included a comparison of the cell proportions in the IMC and FACS in Figure 2. However, it is not clear that this comparison is easy to make from these plots as the FACS results appear to be reported as a proportion to a parent population and the IMC an overall frequency? Further to this, the total cell type counts for the scRNAseq are then reported in Figure 4 making them difficult to compare to the other two technologies.

Reviewer #3 (Remarks to the Author):

The authors carefully revised the manuscript according to the reviewer's comments. They accepted/discussed all the comments and the manuscript has improved substantially after revision. I have no additional comments.

REVIEWERS' COMMENTS

Reviewer #2 (Remarks to the Author):

Since the previous version the authors have expanded their methods section and provided the code needed to perform the analysis in their manuscript. This has improved the reproducibility of their results. In addition to this they have made their IMC data publicly available which will benefit the broader community. To their credit, the authors have substantially altered the way that they preprocessed their IMC data following feedback from the reviewers. This appears to be concordant with community standards for the analysis of this data type.

#1 response to reviewer 2 concerns-Given the improvements in the pre-processing of the IMC data, Figure 1H is very concerning. It is very hard to distinguish how the clusters were annotated as certain cell types given the colours and intensities within the heatmap. For instance, CD4 and CD8 are coexpressed in more than half of the clusters. This is a key figure which should provide confidence in the unsupervised clustering used to define cell types in the experiment. This figure makes it difficult to accept any of the conclusions made that rely on the cell type annotations from the IMC data. Figure S5 reduces this confidence further. Minor points related to Figure 1H, the legend describes the colours as z-scored mean marker expression, it is surprising to see all the values between 0.1 and 0.9. The colours of the cell type frequencies are very difficult to see. It would help if the order of the clusters were consistent between g and h.

Reply: thanks for the critical comments. We understand the concern of Figure 1H and Figure S5. The co-expression of lineage markers are almost impossible to avoid during IMC experiments. Please refer our detailed reply in reviewer#1 question #3 to 6. The limitation is due to the resolution limit of IMC (1 um). The marker co-expression will be more problematic when the cell density is very high, such as the situation of heavy immune cell infiltration in UC region. During our cell annotation, we try to avoid any possible error due to lineage marker co-expression. For example, if both CD4+ and

CD8+ are expressed, we won't define the cluster as CD4+ or CD8+ T cells. Instead, we define the cluster as T cells.

The legend of Figure 1H was wrong in the previous version of manuscript. The normalization style is max-min. We have corrected the error. Please refer to line#873-874.

Thanks for the nice suggestion. We have aligned the order of figure 1g and h.

Minor points.

#2 response to reviewer 1 concerns- T-tests are not appropriate in Figure 2 and Figure 6. I am not sure if this would alter the conclusions substantially though.

Figures were stretched in Figure 3.

Reply: thanks for the careful reading and critical comments. We compared cellular cluster frequency and cellular neighborhood frequency between UC and healthy regions in Figure 2 and 6, respectively. And we replaced the t-test with two-sided Wilcoxon rank-sum tests, which can be used to compare two independent groups of samples. Statistical methods were updated (line#612-620). The conclusion still holds true.

#3 response to reviewer 1 concerns-Font size in Figure 4 in particular is very small. Font sizes across most panels in most figures are very inconsistent. I am sure it would be beneficial for readers if this was addressed. Maybe the number of panels in each figure need to be reduced to address this.

Reply: thanks for the very nice suggestions for figure preparation. We have reduced the size of each panel and increased the font size for better reading experience.

#4 response to reviewer 1 concerns-I appreciate that the authors included a comparison of the cell proportions in the IMC and FACS in Figure 2. However, it is not clear that this comparison is easy to make from these plots as the FACS results appear to be reported as a proportion to a parent population and the IMC an overall frequency? Further to this, the total cell type counts for the scRNAseq are then reported in Figure 4 making

them difficult to compare to the other two technologies.

Reply: thanks for the very nice suggestions for data presentation. We have aligned the comparison standard that total cell counts was used as denominator during specific cell type frequency calculation for different methods such as FACS, IMC and scRNA-seq.

Reviewer #3 (Remarks to the Author):

#1 response to reviewer 3-The authors carefully revised the manuscript according to the reviewer's comments. They accepted/discussed all the comments and the manuscript has improved substantially after revision. I have no additional comments.

Reply: thanks for all the constructive comments and suggestions.